# Synergistic effects of ion transporter and MAP kinase pathway inhibitors in melanoma

Ugur Eskiocak[1], Vijayashree Ramesh[1], Jennifer G. Gill[1,2], Zhiyu Zhao[1], Stacy W. Yuan[1], Meng Wang[1], Travis Vandergriff[2], Mark Shackleton[3,4], Elsa Quintana[5,†], Timothy M. Johnson[6], Ralph J. DeBerardinis[1] & Sean J. Morrison[1,7]

New therapies are required for melanoma. Here, we report that multiple cardiac glycosides, including digitoxin and digoxin, are significantly more toxic to human melanoma cells than normal human cells. This reflects on-target inhibition of the ATP1A1 $Na^+/K^+$ pump, which is highly expressed by melanoma. MEK inhibitor and/or BRAF inhibitor additively or synergistically combined with digitoxin to induce cell death, inhibiting growth of patient-derived melanomas in NSG mice and synergistically extending survival. MEK inhibitor and digitoxin do not induce cell death in human melanocytes or haematopoietic cells in NSG mice. In melanoma, MEK inhibitor reduces ERK phosphorylation, while digitoxin disrupts ion gradients, altering plasma membrane and mitochondrial membrane potentials. MEK inhibitor and digitoxin together cause intracellular acidification, mitochondrial calcium dysregulation and ATP depletion in melanoma cells but not in normal cells. The disruption of ion homoeostasis in cancer cells can thus synergize with targeted agents to promote tumour regression *in vivo*.

[1] Department of Pediatrics, Children's Research Institute, Dallas, Texas 75390, USA. [2] Department of Dermatology, University of Texas Southwestern Medical Center, Dallas, Texas 75390, USA. [3] Cancer Development and Treatment Laboratory, Peter MacCallum Cancer Centre, East Melbourne, Victoria 3002, Australia. [4] Sir Peter MacCallum Department of Oncology and Department of Pathology, University of Melbourne, Parkville, Melbourne, Victoria 3010, Australia. [5] Life Sciences Institute, University of Michigan, Ann Arbor, Michigan 48109-2216, USA. [6] Department of Dermatology, University of Michigan, Ann Arbor, Michigan 48109-2216, USA. [7] Howard Hughes Medical Institute, University of Texas Southwestern Medical Center, Dallas Texas 75390, USA. † Present address: OncoMed Pharmaceuticals, 800 Chesapeake Drive, Redwood City, California 94063, USA. Correspondence and requests for materials should be addressed to S.J.M. (email Sean.Morrison@UTSouthwestern.edu).

The MAPK pathway is activated in 90% of melanomas[1], commonly by gain-of-function mutations in *BRAF* or *NRAS*, but even melanomas lacking these mutations can exhibit pathway activation[2,3]. MAPK pathway inhibition with drugs that inhibit $BRAF^{V600E}$ and/or MEK signalling can induce disease regression and prolong survival of patients with BRAF mutations[4–7]. However, relapses generally occur within months.

We developed a xenograft assay in which melanomas obtained from patients engraft efficiently in NOD/SCID IL2R$\gamma^{null}$ (NSG) mice[8,9]. Melanoma metastasis in this assay is predictive of clinical outcome in patients[10]. Stage III melanomas that metastasize efficiently in NSG mice form distant metastases in patients despite surgical resection, whereas melanomas that metastasize inefficiently in mice are generally cured by surgery in patients[10]. We used this assay to test new therapies.

Cardiac glycosides are toxic to various cancer cell lines in culture, including melanoma cell lines[11,12]. Cardiac glycosides bind and inhibit the α-catalytic subunit of the human $Na^+/K^+$ ATPase encoded by *ATP1A1*, *ATP1A2*, *ATP1A3* or *ATP1A4* (ref. 13). This ATPase pumps $Na^+$ ions out of the cell and $K^+$ ions in to create a $Na+/K+$ gradient that is used by other channels and transporters to transport ions, sugars and amino acids across the plasma membrane. Cardiac glycosides, including digitoxin and digoxin, are widely used in the treatment of heart failure[14]. Inhibition of the $Na^+/K^+$ ATPase depolarizes the plasma membrane of cardiomyocytes, inhibiting $Na^+/Ca^{2+}$ exchangers and leading to the accumulation of intracellular $Ca^{2+}$. This improves cardiomyocyte contractility in the failing heart[15].

Retrospective studies show patients taking cardiac glycosides for a heart indication exhibited a 25% reduction in prostate cancer incidence[16], reduced breast cancer recurrence after mastectomy[17] and better survival outcomes for various carcinomas (breast, colon, liver and head and neck)[18]. Cardiac glycoside use increased the risk of breast cancer or death from prostate cancer in other studies[19–21]. Several phase I and II clinical trials have tested digoxin as a single agent or in combination with chemotherapy, or targeted agents in multiple cancers[22]. These included a phase II trial in melanoma that combined digoxin with cisplatin, IL-2, IFNα and vinblastine[22]. To our knowledge, no results have yet been reported from these trials.

Our data show that single agent activity from cardiac glycosides against melanoma xenografts *in vivo* is limited. However, we find that cardiac glycosides synergize with MAPK pathway inhibitors to extend the survival of mice xenografted with human melanoma or acute myeloid leukaemia cells. The combination of cardiac glycosides and MAPK pathway inhibitors preferentially kill cancer cells *in vivo* by disrupting intracellular pH homoeostasis and dysregulating mitochondrial function.

## Results

**Cardiac glycosides preferentially kill melanoma cells**. We screened 200,000 small molecules for increased toxicity against primary human melanoma cells compared with normal human cells. Multiple cardiac glycosides were more toxic to primary human melanoma cells than to normal human cells, including umbilical cord blood (hUCB) cells and melanocytes (Fig. 1a,b). Addition of low concentrations of digitoxin to culture increased the frequency of activated caspase-3/7$^+$ cells among melanoma cells derived from three patients (M481, M491 and M214), but not normal human melanocytes from two donors (hMEL2 and hMEL3) or immortalized melanocytes from another donor (hiMEL[23]; Supplementary Fig. 1a). The half maximal inhibitory digitoxin concentration (IC$_{50}$) for the human A375 melanoma cell line was 27 nM but for hUCB cells was 22,200 nM (Fig. 1c and Supplementary Fig. 1b). The IC$_{50}$

values for melanoma cells obtained from 8 of 15 patients (15–40 nM; Fig. 1c) fell within the therapeutic range of plasma concentrations used in patients for heart failure (up to 45 nM) (ref. 24). The IC$_{50}$ values for melanoma cells from the other 7 patients (65–1,540 nM) were above the safe therapeutic range. siRNA inhibition of ATP1A1 expression using 3 different siRNAs depleted A375 melanoma cells (Supplementary Fig. 1c,d).

The ability of cardiac glycosides to depolarize the plasma membrane by inhibiting the ATP1A1 $Na^+/K^+$ pump can be measured by staining with the lipophilic dye DiSBAC$_2$(3) (ref. 25), which accumulates and fluoresces in depolarized membranes[26]. Low concentrations of digitoxin induced depolarization in melanomas derived from all three patients but not in most primary human melanocytes (Fig. 1d).

By microarray analysis we detected little or no expression of *ATP1A2*, *ATP1A3* or *ATP1A4* in xenografted melanomas or in normal human melanocytes (Supplementary Fig. 1e,f). However, we observed significantly higher *ATP1A1* expression in all melanomas compared with normal human melanocytes (Fig. 1e). The GEO data sets GDS1375 (ref. 27) and GSE46517 (ref. 28) also showed higher levels of *ATP1A1* expression in primary and metastatic melanoma specimens compared with normal skin and benign nevi (Supplementary Fig. 1g,h).

We tested ATP1A1 protein levels by immunohistochemistry in normal human skin, benign nevi, primary melanoma and metastatic melanoma specimens. ATP1A1 protein was mainly limited to basal keratinocytes in normal human epidermis, but expanded to include melanocytic nests in benign nevi (Supplementary Fig. 1i). In primary and metastatic melanomas, ATP1A1 staining was robust and nearly homogeneous among melanoma cells (Supplementary Fig. 1i). Consistent with our results, immunohistochemistry data from the Human Protein Atlas showed normal skin has limited ATP1A1 expression, while 8 of 10 melanomas have ATP1A1 staining in >75% of cells[29].

Western analysis showed limited ATP1A1 expression in normal human melanocytes, but elevated expression in immortalized melanocytes and some xenografted melanomas (Supplementary Fig. 1k). We observed no correlation between *ATP1A1* mRNA levels and digitoxin IC$_{50}$ values (Supplementary Fig. 1l), *BRAF* mutation status (Supplementary Fig. 1m) or *NRAS* mutation status (Supplementary Fig. 1n).

The on-target effects of cardiac glycosides on human cells can be blocked by expression of mouse *Atp1a1*, which binds cardiac glycosides poorly and is insensitive to therapeutic doses[13]. Ectopic expression of mouse *Atp1a1*, but not human *ATP1A1*, blocked the effects of digitoxin on viability and plasma membrane depolarization in cultured melanoma cells from all three patients (Fig. 1f,g) and A375 melanoma cells (Supplementary Fig. 2a,b). Cardiac glycosides thus kill melanoma cells by inhibiting ATP1A1.

**Cardiac glycosides promote melanoma regression *in vivo***. We subcutaneously xenografted melanomas derived from 11 patients into NSG mice (clinical characteristics shown in Supplementary Fig. 2c). Four of the melanomas had $BRAF^{V600E}$ mutations, three had *NRAS* mutations, three had *NF1* mutations and one had a *MAP2K1* (MEK1) mutation (Fig. 2a). One melanoma was wild-type for *BRAF* and *NRAS* (M214) with an unknown driver mutation (Fig. 2a). After the melanomas formed palpable tumours at the injection site, mice were treated orally with the MEK inhibitor trametinib (0.5 mg kg$^{-1}$ body mass per day) and/or digitoxin (0.5 mg kg$^{-1}$ body mass per day). This digitoxin dose yielded plasma levels in mice of 12 ± 5 nM, well within the human therapeutic range. In all, 4 of 11 melanomas grew significantly more slowly in the presence of digitoxin alone

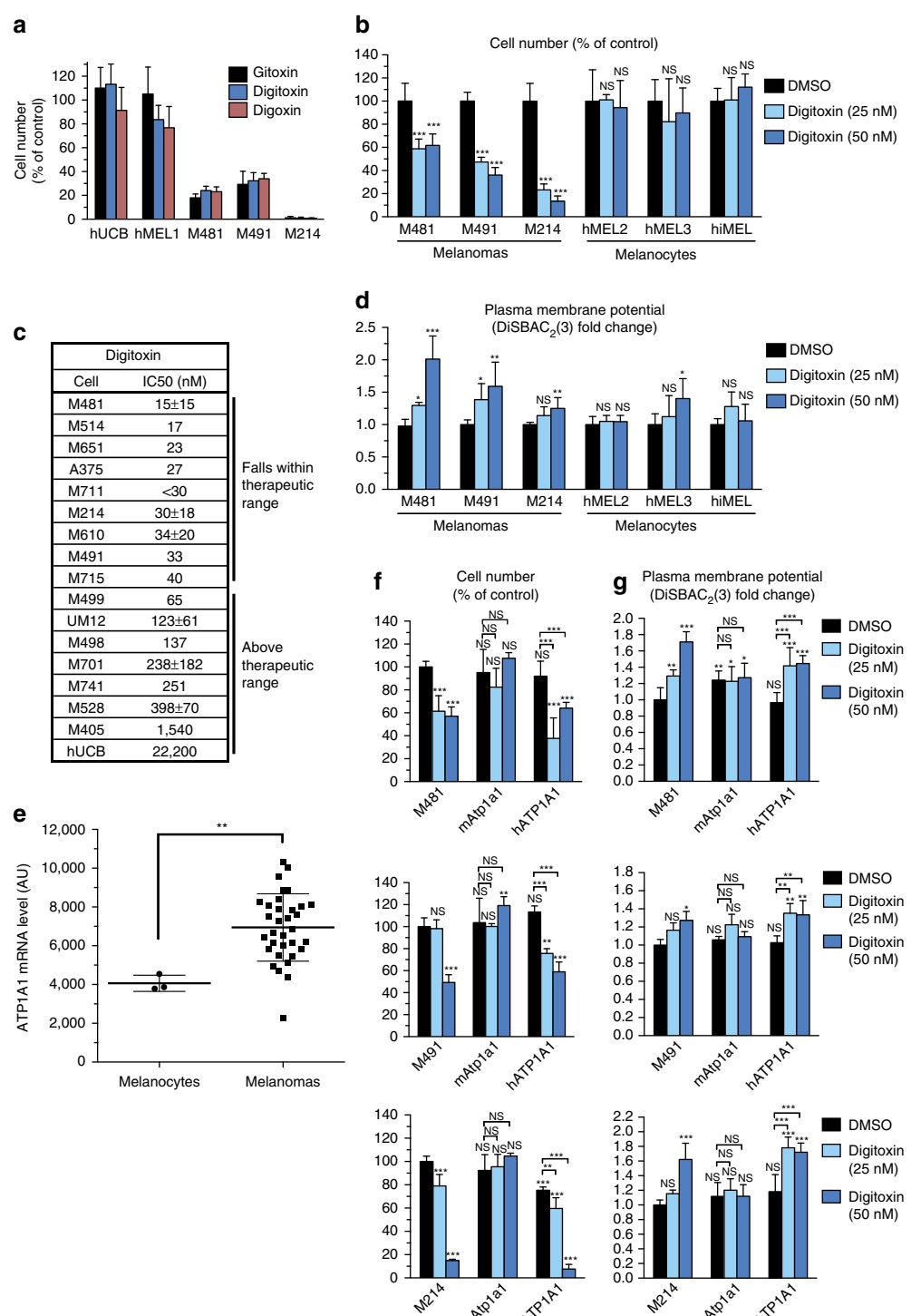

**Figure 1 | Cardiac glycosides are preferentially toxic to melanomas by inhibiting the ATP1A1 Na$^+$/K$^+$ ATPase.** (**a**) Number of hUCB cells, primary melanocytes (hMEL1) or melanoma cells (derived from patients M481, M491 and M214) after 3 days in cultures supplemented with the cardiac glycosides gitoxin (10 μM), digitoxin (2.5 μM) or digoxin (2.5 μM; n = 8 replicate cultures for each cell type from two independent experiments). (**b**) Number of primary melanoma cells or primary melanocytes or immortalized melanocytes (hiMEL) in cultures supplemented with 25 or 50 nM digitoxin for 3 days (n = 6 replicate cultures for each cell type from two independent experiments). (**c**) IC$_{50}$ values for digitoxin in cultures of melanoma cells derived from 15 different patients as well as the A375 melanoma cell line and hUCB cells (n = 1–4 independent experiments per cell type). (**d**) Plasma membrane potential of primary melanoma cells or primary melanocytes or immortalized melanocytes treated with 25 or 50 nM digitoxin for 16 h in culture (n = 6 replicate cultures for each cell type from 2 independent experiments). (**e**) *ATP1A1* transcript levels by microarray analysis of 33 patient-derived melanomas and normal melanocytes (**P < 0.01; two-tailed Student's t-test). Microarray data can be found at the NCBI GEO database under the accession code GSE83583. (**f,g**) Rescue of cell number (**f**) and plasma membrane potential (**g**) in digitoxin-treated melanoma cells after expression of cardiac glycoside-insensitive mouse *Atp1a1*, but not human *ATP1A1* (n = 3–10 replicate cultures per melanoma; each melanoma tested in 1–3 independent experiments). All data represent mean ± s.d. Statistical significance was assessed by one-way (**b,d**) or two-way (**f,g**) analysis of variance followed by Dunnett's multiple comparisons test. For all panels, each treatment was compared with controls (NS, not significant; *P < 0.05; **P < 0.01; ***P < 0.001).

(M610, M405, M214, M715; Fig. 2a). In the presence of MEK inhibitor alone, 9 of 11 melanomas grew significantly more slowly, but only 1 melanoma (M214) was reduced in size relative to the pre-treatment tumour (Fig. 2a). In contrast, melanomas treated with the combination of digitoxin and MEK inhibitor were always smaller, on average, than melanomas treated only with MEK inhibitor and were smaller than pre-treatment tumours in 6 of 11 cases (M481, M487, M610, M214, M715, M660; Fig. 2a). Mice treated with the drug combination had significantly smaller tumours than mice treated only with MEK

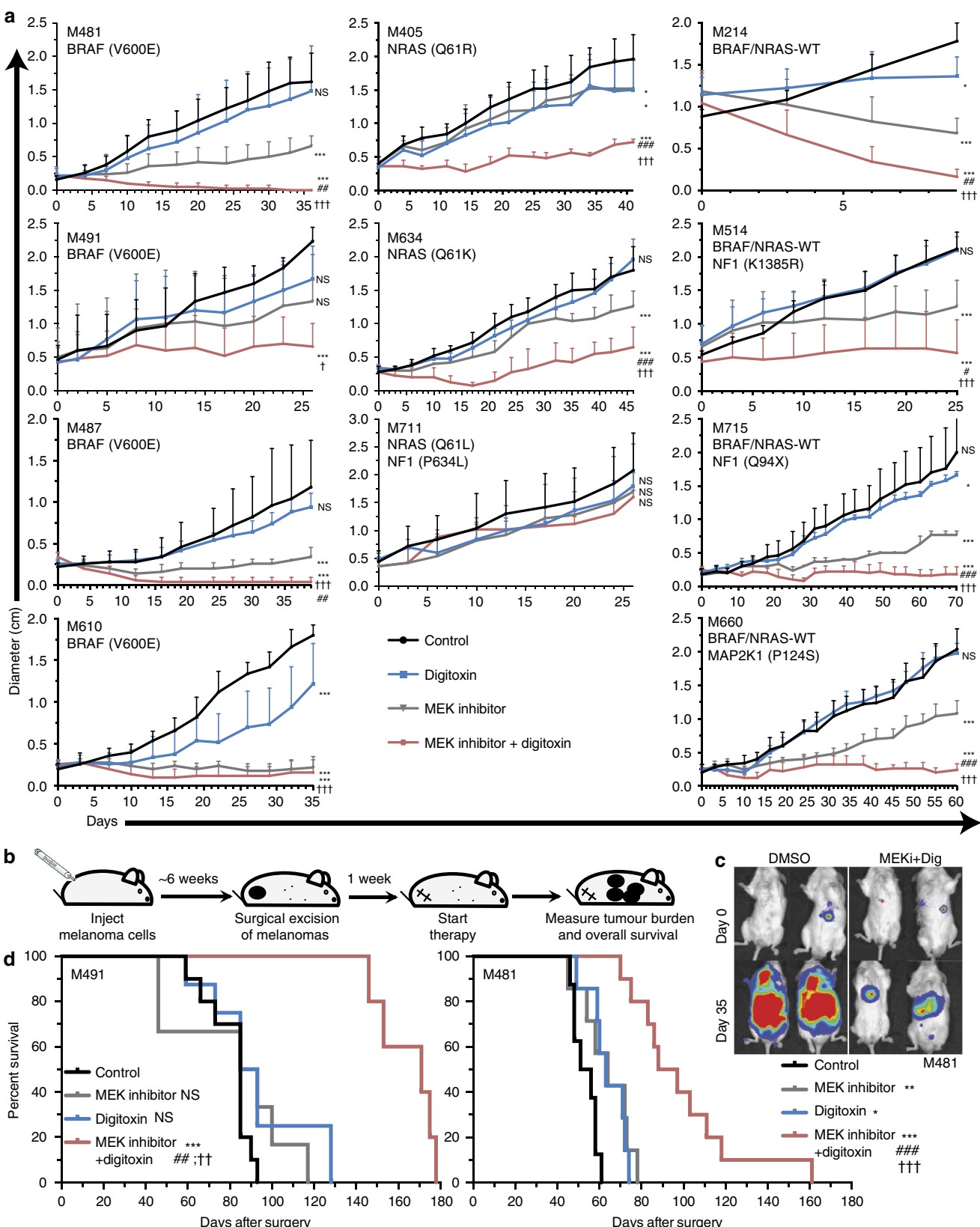

inhibitor or only with digitoxin in 8 of 11 melanomas (M481, M487, M405, M634, M214, M514, M715, M660; Fig. 2a). The combination of digitoxin and MEK inhibitor was therefore significantly more effective than either single agent alone in most xenografts.

Results in Fig. 2a suggest that digitoxin and MEK inhibitor had additive effects on 4 of 11 melanomas (M491, M487, M610 and M214) and synergistic effects on 5 or 6 of 11 melanomas (M481, M634, M514, M715, M660 and perhaps M405).

Similar results were obtained when tumours were treated with digoxin rather than digitoxin (Supplementary Fig. 3a,b) or when the MAPK pathway was inhibited with BRAF inhibitor (vemurafenib) rather than MEK inhibitor (Supplementary Fig. 3c,d). Mice xenografted with a BRAF mutant melanoma treated with digitoxin plus BRAF inhibitor had significantly smaller tumours than mice treated with BRAF inhibitor or digoxin alone.

To test whether digitoxin plus MEK inhibitor would prolong the survival of xenografted mice with metastatic disease we subcutaneously injected melanoma cells derived from two patients (M491, M481), allowed tumours to grow to 2 cm in diameter, then surgically removed the melanomas and initiated treatment with digitoxin and/or MEK inhibitor (Fig. 2b). All mice had disseminated melanoma cells based on the bioluminescence imaging (Fig. 2c, Supplementary Fig. 4b). All untreated mice died within 61 days (M481) or 93 days (M491) of surgery (Fig. 2d). Mice with M491 melanomas treated only with digitoxin or MEK inhibitor did not live significantly longer than control mice, but mice treated with digitoxin plus MEK inhibitor lived significantly longer than mice in all other treatments, up to 178 days after surgery (Fig. 2d). Mice with M481 melanomas treated only with digitoxin or MEK inhibitor lived slightly, but significantly longer than control mice but mice treated with digitoxin plus MEK inhibitor lived significantly longer than mice in all other treatments, up to 161 days after surgery (Fig. 2d). Mice treated with digitoxin plus MEK inhibitor also had a lower tumour burden by bioluminescence imaging (Supplementary Fig. 4b–d).

To test whether the prolonged survival of mice treated with digitoxin plus MEK inhibitor required inhibition of ATP1A1 we over-expressed mouse Atp1a1 in M481 melanoma cells (which is insensitive to therapeutic doses of cardiac glycosides; Supplementary Fig. 5). Expression of mouse Atp1a1 did not significantly affect the survival of untreated mice (Supplementary Fig. 5b). Mice treated with digitoxin plus MEK inhibitor lived significantly longer than untreated mice, but the expression of mouse Atp1a1 partially—but significantly—blocked the increased survival (Supplementary Fig. 5b). On the basis of the bioluminescence imaging, tumours expressing mAtp1a1 grew significantly more quickly than tumours lacking mAtp1a1 in mice treated with digitoxin plus MEK inhibitor (Supplementary Fig. 5c). The combination of digitoxin and MEK inhibitor thus synergistically extended the survival of mice with metastatic melanoma in a manner that depended on human ATP1A1.

We treated mice bearing M214 melanomas with digitoxin and/or MEK inhibitor for 27 days then stopped all therapy.

Digitoxin or MEK inhibitor alone each slowed the growth of tumours, but did not cause tumour regression and the mice had to be killed soon after stopping treatment (Supplementary Fig. 6a). In contrast, the combination of digitoxin plus MEK inhibitor caused tumour regression to the point of undetectability and the tumours did not grow back within 132 days of stopping treatment (Supplementary Fig. 6a).

**Digitoxin plus MEK inhibitor preferentially kills melanoma.** In culture, cardiac glycosides and MEK inhibitor each exhibited activity against human melanoma cells but the effects of the drug combination varied under different culture conditions. Therefore, we studied the mechanism by which digitoxin and MEK inhibitor synergistically promoted the survival of mice with metastatic melanomas in vivo. We subcutaneously xenografted melanomas derived from three patients, waited for tumours to grow to 0.5–0.7 cm in diameter, then treated with digitoxin and/or MEK inhibitor for 4 days. Treatment with digitoxin or MEK inhibitor alone induced cell death (TUNEL stained cells in sections) beyond that observed in control tumours in xenografts derived from only 1 of 3 patients (M214, Fig. 3a). Digitoxin plus MEK inhibitor induced significantly more cell death than in all 3 untreated melanomas and significantly more cell death than digitoxin alone or MEK inhibitor alone in melanomas from 2 of 3 patients (Fig. 3a). Digitoxin and MEK inhibitor thus synergistically induced cell death in melanoma xenografts derived from 2 of 3 patients.

Cell death appeared to occur by apoptosis as digitoxin plus MEK inhibitor significantly increased the number of activated caspase-3$^+$ cells in control and tRFP-expressing melanomas, but not in BCL2 over-expressing melanomas (Fig. 3b). In contrast, we did not detect any clear effect on autophagy as digitoxin and/or MEK inhibitor did not consistently alter p62 or LC3-I/II levels in xenografted melanomas (Supplementary Fig. 6b). It remains possible that digitoxin and MEK inhibitor exert effects on autophagy and/or necrosis in some melanomas.

To assess whether digitoxin plus MEK inhibitor is preferentially toxic to cancer cells we transplanted mice with gelatin sponges containing either human melanoma cells (M214) or immortalized human melanocytes (hiMEL) (one of each type of sponge per mouse, implanted subcutaneously on different flanks) then treated the mice with digitoxin plus MEK inhibitor (Fig. 3c). Digitoxin plus MEK inhibitor induced cell death in melanoma cells but not in immortalized melanocytes (Fig. 3d). We also transplanted hUCBs into NSG mice then treated with digitoxin plus MEK inhibitor (Fig. 3e). Digitoxin plus MEK inhibitor did not have any effect on the frequency of human haematopoietic cells in the bone marrow or blood (Fig. 3f) or the total number of human cells per femur (Fig. 3g). The combination did not induce apoptosis in human or mouse haematopoietic cells (mouse cells were a negative control as they should be insensitive to digitoxin) from the bone marrow or blood (Fig. 3h,i). The drug combination

**Figure 2 | Digitoxin and MEK inhibitor additively or synergistically impair tumour growth and prolong survival of mice with metastatic human melanomas.** (**a**) NSG mice with subcutaneous xenografts derived from 11 different patients were treated daily with digitoxin and/or MEK inhibitor to examine the effect on tumour diameter (mean ± s.d.; n = 3–5 mice per treatment per melanoma; each melanoma tested in an independent experiment). Statistical significance was assessed by two-way analysis of variance followed by Dunnett's multiple comparisons test. (**b**) Schematic of experiments to test the effect of digitoxin and/or MEK inhibitor on the survival of mice with disseminated melanoma. (**c**) Bioluminescence signal from mice engrafted with luciferase-expressing patient-derived xenografts after surgical excision of the primary tumour (day 0) or 35 days later. Bioluminescence data for all mice engrafted with M481 are shown in Supplementary Fig. 4b. (**d**) Survival (not requiring killing per animal care protocol) in days after surgical removal of the primary subcutaneous melanoma (n = 5–10 mice per treatment for M491 and n = 7–10 mice per treatment for M481 in independent experiments). Statistical significance was assessed using the log-rank test. The statistical significance of each treatment compared with control (*P < 0.05; **P < 0.01; ***P < 0.001), the combination compared with MEK inhibitor alone ($^\#P < 0.05$; $^{\#\#}P < 0.01$; $^{\#\#\#}P < 0.001$) or the combination compared with digitoxin alone ($^\dagger P < 0.01$; $^{\dagger\dagger}P < 0.01$; $^{\dagger\dagger\dagger}P < 0.001$).

is thus preferentially toxic to melanomas as compared with human melanocytes and haematopoietic cells *in vivo*.

**Mechanism by which digitoxin and MEK inhibitor act**. MEK inhibitor significantly reduced ERK phosphorylation (pERK), as

expected, in melanomas from all three patients, including *BRAF*[V600E] and *BRAF* wild-type melanomas (Fig. 3j, Supplementary Fig. 8). In contrast, levels of pERK did not differ between digitoxin-treated and control melanomas (Fig. 3j). The combination of digitoxin and MEK inhibitor reduced pERK levels to a similar extent as MEK inhibitor alone.

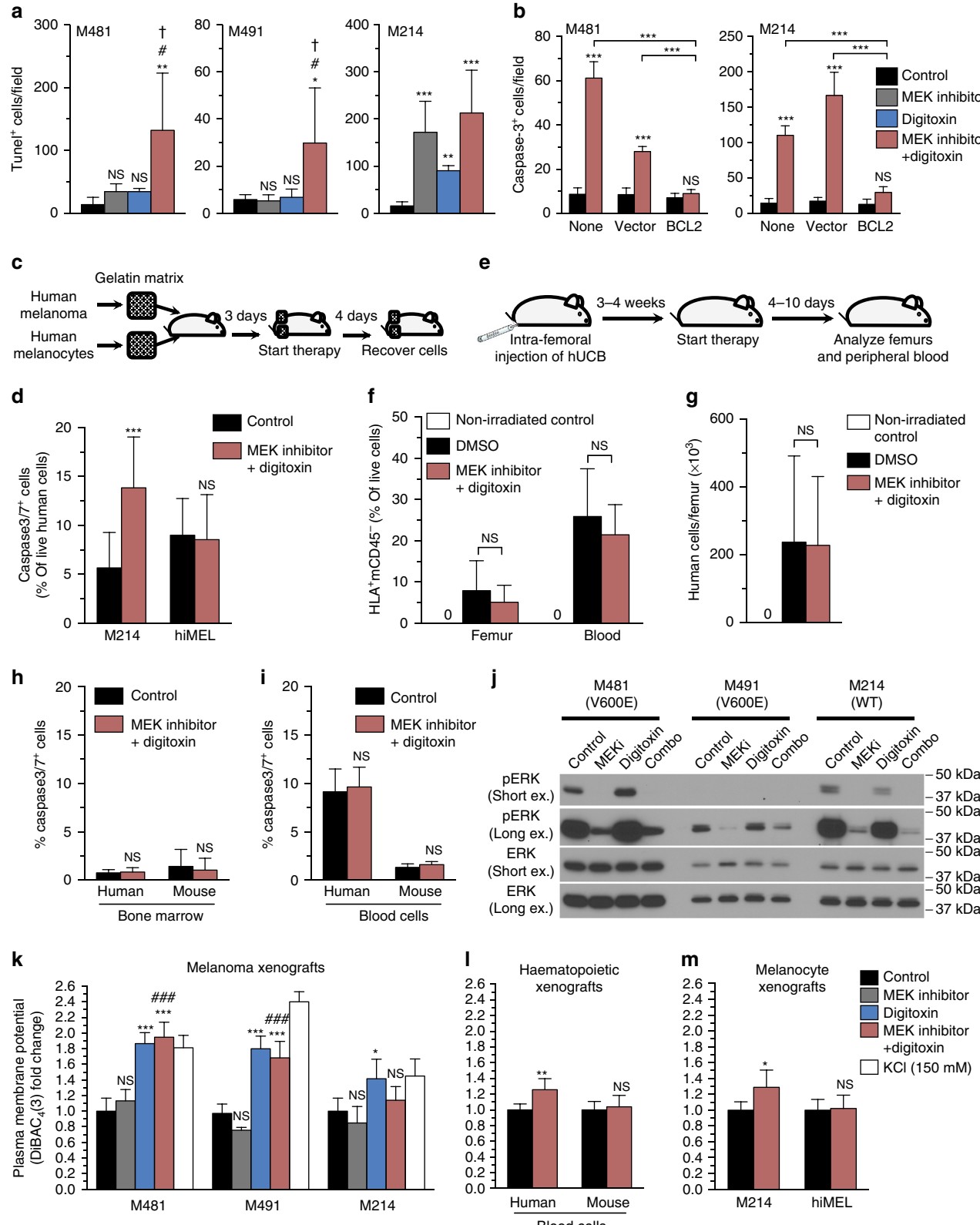

We measured plasma membrane potential using DiBAC$_4$(3), which like DiSBAC$_2$(3) (Fig. 1g) inserts and fluoresces in depolarized plasma membranes[26] and has been used to monitor the effects of cardiac glycosides on membrane depolarization[30,31]. Treatment of xenografted mice with digitoxin depolarized the plasma membranes of melanoma cells from all three patients (Fig. 3k). MEK inhibitor did not significantly affect membrane potential. Treatment with the combination of digitoxin and MEK inhibitor had effects similar to digitoxin alone (Fig. 3k). The two compounds thus did not have additive or synergistic effects on MEK signalling or plasma membrane potential.

Digitoxin or digitoxin plus MEK inhibitor also depolarized the plasma membranes of some melanocyte lines in culture (Fig. 1d) and xenografted human haematopoietic cells (Fig. 3l) but not immortalized melanocytes (Fig. 3m). Thus, the increased toxicity of cardiac glycosides to melanoma cells did not correlate with a difference in the ability of cardiac glycosides to depolarize the plasma membranes of melanoma cells as compared with normal cells.

Consistent with earlier reports[32,33], MEK inhibition tended to reduce glucose uptake by melanoma cells but treatment with digitoxin did not accentuate this effect (Supplementary Fig. 7a). To assess the effects of digitoxin and MEK inhibitor on glucose metabolism, we administered uniformly $^{13}$C labelled glucose to NSG mice bearing melanomas derived from two patients. Treatment with MEK inhibitor did not significantly affect the fractional enrichment of labelled glucose in tumours but tended to reduce the fractional contribution of labelled glucose to the downstream pools of lactate, citrate and glutamate (Supplementary Fig. 7b–d). Digitoxin did not significantly affect the fractional contribution of labelled glucose to the downstream pools of lactate, citrate and glutamate, either by itself or in combination with MEK inhibitor (Supplementary Fig. 7b–d). Thus digitoxin plus MEK inhibitor did not appear to have additive or synergistic effects on glucose metabolism.

MEK inhibition also reduced cell size, consistent with potential effects on cytoplasm osmolarity, mTOR signalling or cell cycle, but digitoxin did not consistently affect cell size, and the combination usually did not differ from MEK inhibitor alone (Supplementary Fig. 7e).

Cardiac glycosides can inhibit HIF1α translation when used at concentrations that are much higher than those used in our experiments[34]. We detected inhibition of HIF1α expression at micromolar concentrations of digitoxin but not at the concentrations used in our experiments (Supplementary Fig. 7f), suggesting HIF1α inhibition does not explain our results.

**Digitoxin plus MEK inhibitor induces cellular acidification.** The disruption of Na$^+$/K$^+$ gradients by cardiac glycosides and the inhibition of ERK signalling by MEK inhibitor can each inhibit Na$^+$/H$^+$ exchangers (NHEs), either directly or by modulating p90RSK function, promoting cytoplasmic acidification[35–37]. NHE1 was strongly expressed by all melanomas, although lower expression of other NHE family members was observed in subsets of melanomas. Normal cells typically have an intracellular pH of 6.9 to 7.1 while cancer cells typically have higher and more variable intracellular pHs (7.2 to 7.7)[38]. We used SNARF1, a ratiometric pH-sensitive indicator[39], to investigate the effect of MEK inhibitor and/or digitoxin on the intracellular pH of xenografted melanomas. In melanomas from all three patients, MEK inhibitor alone did not significantly affect intracellular pH but treatment with the drug combination significantly reduced intracellular pH (Fig. 4a). Treatment with digitoxin alone significantly reduced intracellular pH in 1 of 3 melanomas, but to a significantly lesser extent than the drug combination. Digitoxin and MEK inhibitor thus have additive or synergistic effects on cytoplasmic acidification in melanoma cells *in vivo*. In contrast, digitoxin plus MEK inhibitor did not significantly affect the intracellular pH of human haematopoietic cells (Fig. 4b) or human melanocytes (Fig. 4c) in NSG mice.

To test whether digitoxin plus MEK inhibitor caused intracellular acidification by inhibiting NHE function we assessed the effect of digitoxin and/or MEK inhibitor on NHE activity in M481 and M214 cells (Fig. 4d). We used a kinetic flow cytometry assay in which recovery from intracellular acidification induced by propionic acid is dependent on NHE activity and is measured as $\Delta$pH s$^{-1}$ (ref. 40). Treatment with MEK inhibitor or digitoxin alone did not significantly affect NHE activity in either melanoma (Fig. 4d). In contrast, treatment with digitoxin plus MEK inhibitor significantly reduced NHE activity in both melanomas compared with control mice and mice treated with digitoxin or MEK inhibitor alone (Fig. 4d). Digitoxin and MEK inhibitor thus synergistically impaired NHE activity in melanoma cells.

To test whether NHE inhibition could explain the effects of digitoxin plus MEK inhibitor on melanoma cells we treated mice xenografted with M481 and M214 melanomas with amiloride, an inhibitor of Na$^+$ channels and Na$^+$-dependent ion transporters including NHE[41–43]. Amiloride treatment (20 mg kg$^{-1}$ body mass per day) phenocopied the effects of digitoxin plus MEK inhibitor, significantly reducing intracellular pH (Fig. 4e) and increasing cell death (Fig. 4f) in both melanoma lines, although not to the extent of digitoxin plus MEK inhibitor.

**Figure 3 | MEK inhibitor reduces ERK phosphorylation while digitoxin depolarizes the plasma membrane and the combination additively or synergistically promotes cell death.** (**a,b**) Patient-derived xenografts were treated 4 days *in vivo* with MEK inhibitor and/or digitoxin then surgically excised, fixed and sectioned. (**a**) Cell death by TUNEL ($n = 3$–5 mice per treatment per melanoma; each melanoma tested in independent experiments). (**b**) Melanomas (M481, M214) infected with BCL2 or control construct (turbo RFP) that had formed tumours in the flanks of NSG mice, were treated 4 days with digitoxin plus MEK inhibitor and stained for activated caspase-3$^+$ ($n = 4$ mice per treatment; each melanoma tested in independent experiments; statistical significance was assessed with one-way analysis of variance followed by Dunnett's multiple comparisons test). (**c,d**) Mice were subcutaneously transplanted with gelatin sponges containing M214 melanoma cells or immortalized melanocytes (hiMEL) in opposite flanks then treated for four days with digitoxin plus MEK inhibitor ($n = 14$ sponges per treatment from three independent experiments; statistical significance was assessed by unpaired two-tailed $t$-tests). (**e–i**) Mice engrafted with human (HLA$^+$) hUCB cells were treated for 4–10 days with digitoxin plus MEK inhibitor ($n = 7$ mice per treatment from two independent experiments; statistical significance was assessed by unpaired two-tailed $t$-tests). (**j**) Western blot analyses of xenografts treated for 4 days *in vivo* with MEK inhibitor and/or digitoxin (full blot in Supplementary Fig. 8). (**k**) Relative DiBAC$_4$(3) fluorescence, measuring plasma membrane potential in live melanoma cells acutely dissociated from tumours treated for 4 days *in vivo* ($n = 4$–5 mice per treatment per melanoma; each melanoma tested in independent experiments; statistical significance was assessed by one-way analysis of variance followed by Dunnett's multiple comparisons test). Dissociated cells treated with 150 mM KCl were a positive control for depolarization. (**l**) Plasma membrane potential of haematopoietic cells in the blood of xenografted NSG mice treated with digitoxin plus MEK inhibitor for 10 days ($n = 5$ mice per treatment; statistical significance was assessed by unpaired two-tailed $t$-tests). (**m**) Plasma membrane potential of hiMEL melanocytes or M214 melanoma cells in gelatin sponges from opposing flanks of mice treated with digitoxin plus MEK inhibitor for 4 days ($n = 4$ mice per treatment; statistical significance assessed by two-tailed $t$-tests). All data represent mean ± s.d. For all statistical tests, comparisons with control (NS, not significant; *$P < 0.05$; **$P < 0.01$; ***$P < 0.001$) or the combination compared with MEK inhibitor alone ($^{\#}P < 0.05$; $^{\#\#\#}P < 0.001$) or the combination compared with digitoxin alone ($^{\dagger}P < 0.05$; $^{\dagger\dagger}P < 0.01$; $^{\dagger\dagger\dagger}P < 0.001$).

To more directly test whether NHE mediates the effects of digitoxin plus MEK inhibitor on melanoma cells we over-expressed NHE1 in melanoma cells. Over-expression of NHE1 partially but significantly rescued the effects of digitoxin plus MEK inhibitor on intracellular pH (Fig. 4g) and cell death (Fig. 4h). In the absence of digitoxin plus MEK inhibitor, NHE1 over-expression did not significantly affect intracellular pH or cell death (Fig. 4g,h). These results suggest digitoxin plus MEK inhibitor kill melanoma cells partly by synergistically inhibiting NHE function, leading to intracellular acidification.

BCL2 over-expression blocked the effects of digitoxin plus MEK inhibitor on melanoma cell death (Fig. 3b) but did not

rescue the effects on intracellular acidification (Fig. 4i). This indicates BCL2 acted downstream of the effects of digitoxin plus MEK inhibitor on intracellular pH to prevent cell death.

**Digitoxin and MEK inhibitor dysregulate mitochondrial $Ca^{2+}$.** Intracellular acidification increases intracellular $Ca^{2+}$ levels by inhibiting various $Ca^{2+}$ transporters[44,45]. Cardiac glycosides can also increase intracellular $Ca^{2+}$ levels by indirectly inhibiting $Na^+/Ca^{2+}$ exchangers[15]. This raised the possibility that digitoxin and MEK inhibitor might additively or synergistically increase intracellular $Ca^{2+}$ levels in melanoma cells. We first

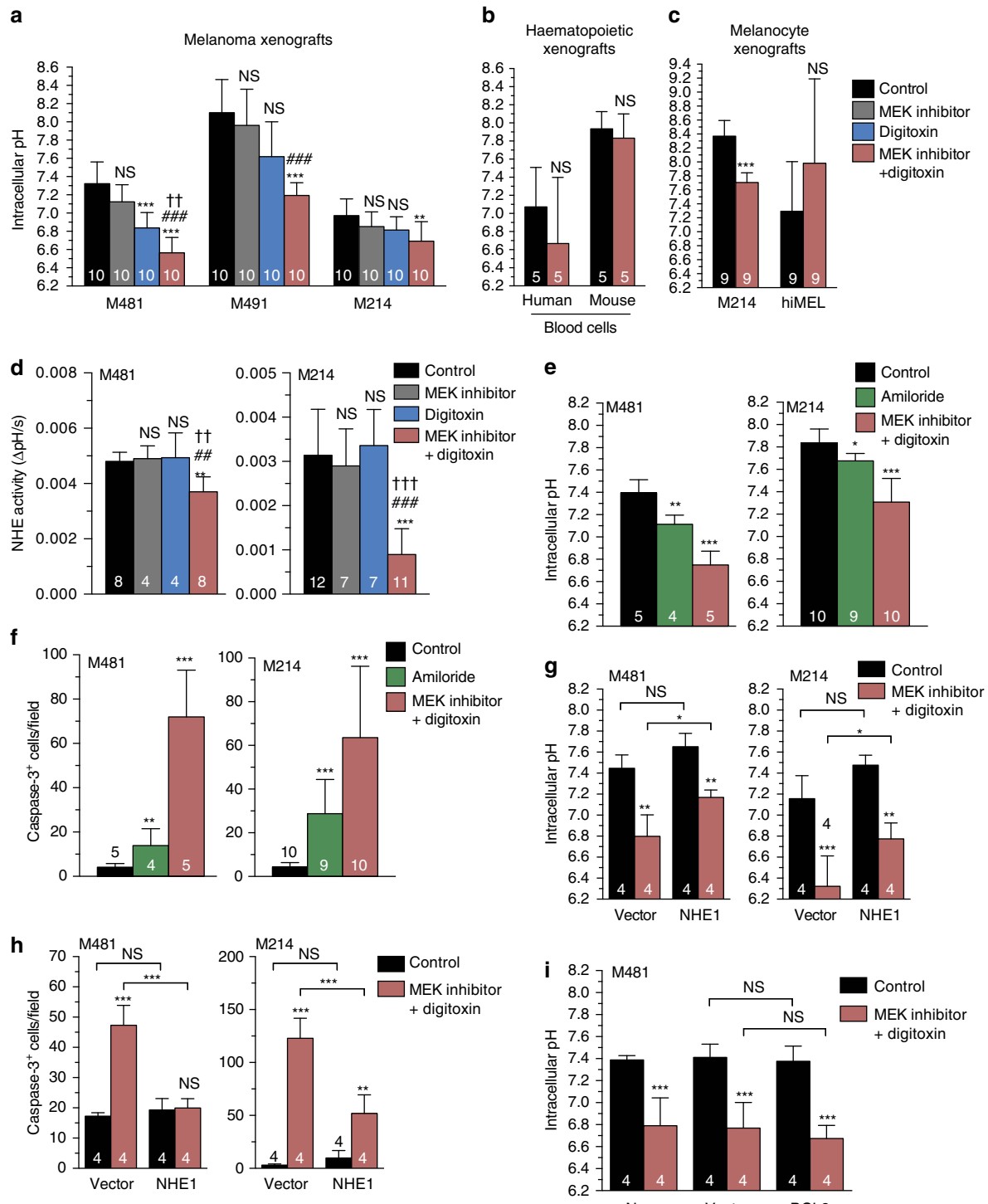

assessed this by measuring cytoplasmic $Ca^{2+}$ based on the Fluo3 fluorescence[46]. We did not detect any effect of digoxin and/or MEK inhibitor on cytoplasmic $Ca^{2+}$ levels (Fig. 5a), possibly reflecting rapid sequestration of cytoplasmic $Ca^{2+}$ into endoplasmic reticulum and/or mitochondria[47,48].

To measure $Ca^{2+}$ levels in mitochondria we adherently cultured melanoma cells from three patients or the A375 melanoma cell line, and stained with Rhod-2, which partitions into mitochondria and fluoresces with increasing $Ca^{2+}$ concentration[49]. We confirmed mitochondrial localization of Rhod-2 by co-staining with Mitotracker deep red (Supplementary Fig. 7g). Addition of digitoxin or MEK inhibitor alone to cultures only modestly increased mitochondrial $Ca^{2+}$ levels (Fig. 5b–k). In contrast, digitoxin plus MEK inhibitor significantly increased mitochondrial $Ca^{2+}$ levels relative to control cells or cells treated with digitoxin alone (except for M491) or MEK inhibitor alone (Fig. 5b–k). Digitoxin and/or MEK inhibitor did not significantly affect mitochondrial $Ca^{2+}$ levels in cultured human haematopoietic cells (Fig. 5g) or melanocytes (Fig. 5h,i). Digitoxin and MEK inhibitor thus synergistically increased mitochondrial $Ca^{2+}$ levels in human melanoma cells but not in normal human cells.

Consistent with evidence that intracellular acidification in response to digitoxin plus MEK inhibitor was at least partially mediated by reduced NHE function (Fig. 4d,g), the increase in mitochondrial $Ca^{2+}$ levels in melanoma cells in response to digitoxin plus MEK inhibitor was also partially rescued by increased NHE1 expression (Fig. 5j,k). NHE1 over-expression by itself did not appear to affect mitochondrial $Ca^{2+}$ levels in melanoma cells not treated with digitoxin plus MEK inhibitor (Fig. 5h,i). Altogether, our results indicate that digitoxin and MEK inhibitor synergistically inhibit NHE function (Fig. 4d), leading to intracellular acidification (Fig. 4a), dysregulated mitochondrial $Ca^{2+}$ levels (Fig. 5b–k) and cell death (Fig. 4f,h) in melanoma cells.

To test whether the increase in mitochondrial $Ca^{2+}$ levels in melanoma cells treated with digitoxin plus MEK inhibitor contributed to the death of melanoma cells, we tested whether BAPTA-AM (an intracellular calcium chelator) reduced cell death induced in melanoma xenografts. One limitation was that BAPTA treatment of xenografted mice, by itself, induced cell death among melanoma cells. In melanomas derived from two patients, this made it impossible to test whether BAPTA rescued the effects of digitoxin plus MEK inhibitor. However, melanoma M214 better tolerated BAPTA treatment. In this melanoma, BAPTA treatment blocked the increase in cell death induced by digitoxin plus MEK inhibitor (Fig. 5l) without affecting the intracellular acidification (Fig. 5m). These data suggest the increase in mitochondrial $Ca^{2+}$ levels contributes to melanoma cell death caused by digitoxin plus MEK inhibitor but that the increase in mitochondrial $Ca^{2+}$ levels occurs downstream of the intracellular acidification.

**Digitoxin and MEK inhibitor impair mitochondrial function.** Since a sustained increase in mitochondrial $Ca^{2+}$ leads to mitochondrial depolarization and dysfunction[47,48], we assessed mitochondrial function in melanoma cells. We first examined mitochondrial membrane potential ($\Delta\Psi_m$) in xenografted melanomas (Fig. 6a). We acutely stained dissociated melanoma cells with MitoTracker deep red, which accumulates in active mitochondria in a $\Delta\Psi_m$ dependent manner[50]. Treatment with MEK inhibitor alone significantly reduced $\Delta\Psi_m$ in 1 of 3 melanomas (M481). Digitoxin alone significantly reduced $\Delta\Psi_m$ in 2 of 3 melanomas (M481, M491). Digitoxin plus MEK inhibitor significantly reduced $\Delta\Psi_m$ in melanomas from all three patients relative to control tumours and tumours treated only with MEK inhibitor (Fig. 6a). Digitoxin plus MEK inhibitor did not significantly affect mitochondrial membrane potential in normal human haematopoietic cells (Fig. 6b) or melanocytes (Fig. 6c) in NSG mice. Digitoxin and MEK inhibitor thus have additive or synergistic effects on mitochondrial membrane potential in melanoma cells but not in normal haematopoietic cells or melanocytes.

Mitochondria are a major source of reactive oxygen species (ROS) and reduced mitochondrial activity can reduce ROS levels[51]. We therefore assessed ROS levels in melanoma cells by staining acutely dissociated melanoma cells from the same tumours with 2′–7′-dichlorofluorescein diacetate (DCFDA), which fluoresces on oxidation[52], and assessed fluorescence in live cells by flow cytometry (Fig. 6d). Treatment with MEK inhibitor or digitoxin alone significantly reduced ROS levels in xenografted melanomas from all three patients relative to controls. Treatment with digitoxin plus MEK inhibitor significantly reduced ROS levels in melanomas from all three patients relative to control tumours or tumours treated only with digitoxin or MEK inhibitor (in 2 of 3 melanomas) (Fig. 6d). In contrast, digitoxin plus MEK inhibitor did not significantly affect ROS levels in normal human haematopoietic cells (Fig. 6e) or melanocytes (Fig. 6f) in NSG mice. Digitoxin and MEK inhibitor thus have additive negative effects on ROS generation within melanoma cells but not in normal haematopoietic cells or melanocytes.

We measured $NAD^+$ and ATP levels within melanoma cells in the same experiments. We sorted live melanoma cells from each tumour and measured $NAD^+$ levels using a colorimetric enzymatic assay[53]. Treatment with MEK inhibitor or digitoxin alone did not significantly affect $NAD^+$ levels in any of the melanomas (Fig. 6g). Treatment with digitoxin plus MEK inhibitor significantly reduced $NAD^+$ levels in melanomas from all 3 patients compared with control mice, significantly more than MEK inhibitor alone in 2 out of 3 melanomas, and significantly more than digitoxin alone in 2 out of 3 melanomas

**Figure 4 | Digitoxin and MEK inhibitor additively or synergistically reduced intracellular pH in melanoma cells by synergistically inhibiting the NHE $Na^+/H^+$ exchanger.** (**a**) Intracellular pH in live melanoma cells acutely dissociated from tumours treated for 4 days in vivo (each melanoma was tested in two independent experiments). (**b**) Intracellular pH of human and mouse cells in the blood of NSG mice xenografted with hUCB cells and treated with digitoxin plus MEK inhibitor for 10 days. (**c**) Intracellular pH of immortalized melanocytes (hiMEL) and M214 melanoma cells grown subcutaneously in gelatin sponges in NSG mice that were treated with digitoxin plus MEK inhibitor for four days ($n = 2$ independent experiments). (**d**) NHE activity in dissociated tumour cells isolated from mice treated 4 days in vivo with digitoxin and/or MEK inhibitor was measured based on their rate of recovery from sodium propionate-induced acute intracellular acidification (each melanoma was tested in 2–3 independent experiments). (**e,f**) Intracellular pH in dissociated cells (**e**) and activated caspase-3$^+$ cells in sections (**f**) from melanomas (M481 and M214) obtained from mice treated for 4 days with amiloride (NHE inhibitor) or digitoxin plus MEK inhibitor ($n = 4$–9 mice per treatment; each melanoma was tested in 1–2 independent experiments). In these experiments, an earlier passage of M214 cells was used, with a higher intracellular pH, compared with the M214 cells shown in **a** and **g**. (**g,h**) Intracellular pH in dissociated tumour cells (**g**) and activated caspase-3$^+$ cells in sections (**h**) of melanoma xenografts (M481 and M214) expressing vector (tRFP control) or NHE1 and treated 4 days with digitoxin plus MEK inhibitor (each melanoma was tested in an independent experiment). (**i**) Intracellular pH in dissociated tumour cells from M481 xenografts that were uninfected, expressing vector (tRFP control) or BCL2 and treated 4 days with digitoxin plus MEK inhibitor. Statistical significance was assessed by one-way analysis of variance followed by Dunnett's multiple comparisons test. All data represent mean ± s.d. In each panel the number of mice per treatment is written on the bars.

(Fig. 6g). Digitoxin and MEK inhibitor thus additively or synergistically reduced NAD[+] levels within melanoma cells.

We sorted live melanoma cells from each tumour and measured ATP levels using a luciferase-based assay[54]. Treatment with MEK inhibitor alone or digitoxin alone did not significantly affect ATP levels in any melanomas (Fig. 6h). In contrast, treatment with digitoxin plus MEK inhibitor significantly reduced ATP levels in all three melanomas compared with control mice or mice treated only with digitoxin, and significantly more than MEK inhibitor alone in 2 out of 3 melanomas (Fig. 6h). Digitoxin plus MEK inhibitor did not significantly affect ATP levels in normal human

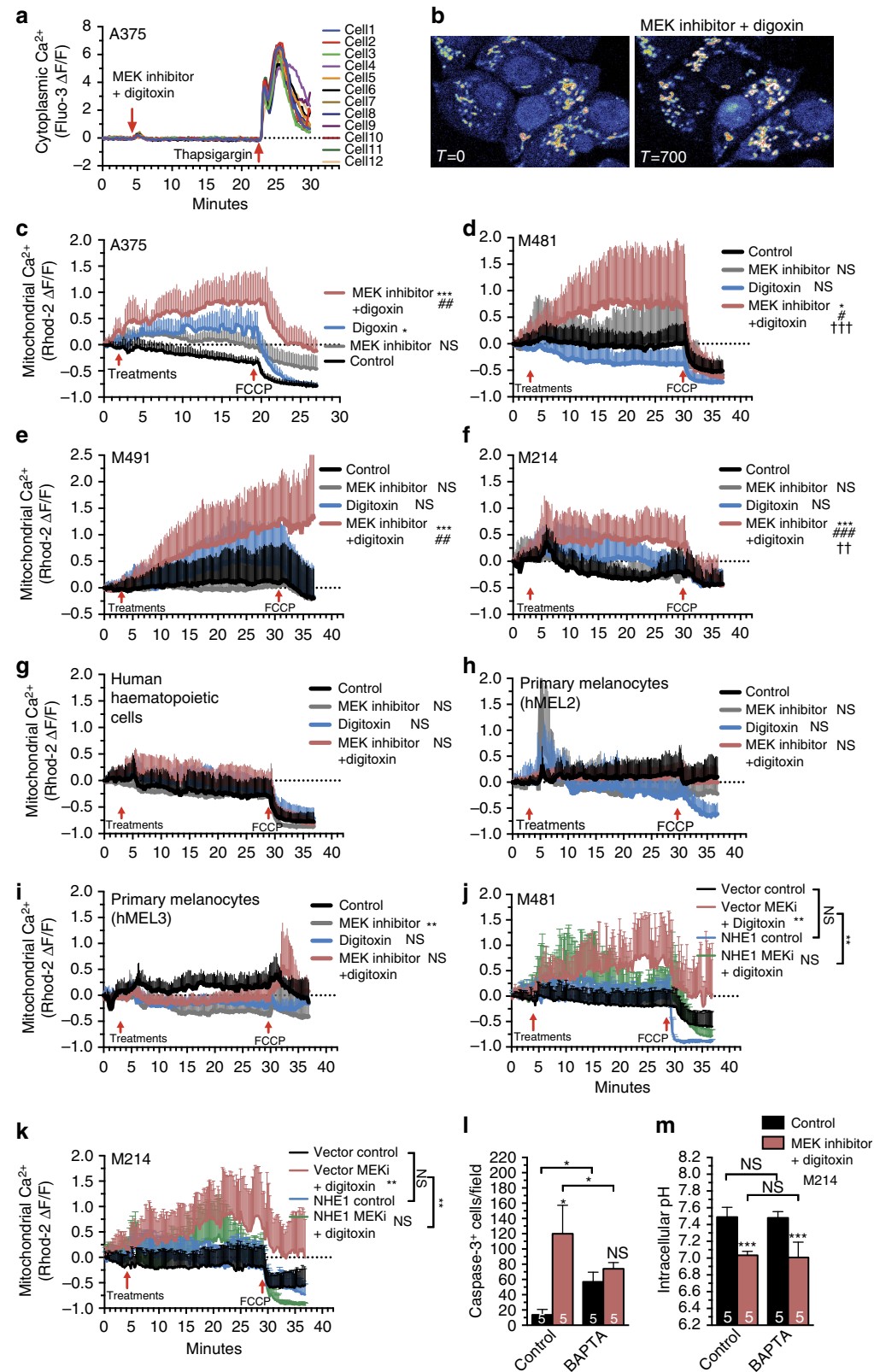

haematopoietic cells (Fig. 6i) or in human melanocytes (Fig. 6j) in NSG mice. Digitoxin and MEK inhibitor thus synergistically reduced ATP levels within melanoma cells but not in normal human haematopoietic cells or melanocytes.

**Digitoxin and MEK inhibitor inhibit NHE function**. Taken altogether, our data suggest that digitoxin plus MEK inhibitor synergistically inhibits NHE function (Fig. 4d), leading to intracellular acidification (Fig. 4a), synergistically increased mitochondrial $Ca^{2+}$ levels (Fig. 5c–f), reduced mitochondrial function (Fig. 6a,d), synergistically depleted ATP (Fig. 6h) and additively or synergistically increased cell death (Fig. 3a; summary in Fig. 7a). Given that NHE1 over-expression partially rescued the effects of digitoxin plus MEK inhibitor on intracellular acidification (Fig. 4g), mitochondrial $Ca^{2+}$ levels (Fig. 5j,k) and cell death (Fig. 4h), our results raised the question of whether NHE1 over-expression could rescue the effects of digitoxin plus MEK inhibitor on tumour growth. NHE1 over-expression by itself had little (M481) or no (M214) effect on the growth of xenografted melanomas (Fig. 7b,c). However, in both melanomas NHE1 over-expression significantly but partially rescued the effects of digitoxin plus MEK inhibitor on tumour growth (Fig. 7b,c). These results are consistent with those above in demonstrating that the therapeutic effects of digitoxin plus MEK inhibitor on melanoma growth are mediated partly by effects on NHE function.

**Digitoxin plus BRAF and MEK inhibitors**. We tested whether the addition of digoxin would enhance responses of xenografted melanomas to BRAF inhibitor (dabrafenib) plus MEK inhibitor (trametinib). BRAF plus MEK inhibitor significantly slowed the growth of BRAF mutant melanomas derived from four patients (M481, M491, M610, M528; Fig. 8a). Addition of digoxin to this drug combination significantly enhanced the treatment response of all melanomas, halting or nearly halting the growth of most melanomas (Fig. 8a).

**Digitoxin plus MEK inhibitor inhibits AML**. To assess whether melanomas are uniquely sensitive to the synergistic effects of cardiac glycosides and MAPK pathway inhibitors we transplanted human NRAS mutant acute myeloid leukaemia cell lines (HL60 and U937) into NSG mice and monitored disease progression with bioluminescence imaging. We initiated treatment with MEK inhibitor and/or digitoxin when the bioluminescence signal became detectable in mice 1–2 weeks after transplantation. Treatment with digitoxin or MEK inhibitor alone each significantly slowed the growth of HL60 cells (Fig. 8b) but not U937 cells (Fig. 8d). Treatment with digitoxin alone did not significantly affect the survival of mice transplanted with either leukaemia (Fig. 8c,e). Treatment with MEK inhibitor alone slightly but significantly extended the survival of mice

transplanted with HL60 cells (Fig. 8c) but not U937 cells (Fig. 8e). In contrast, treatment with digitoxin plus MEK inhibitor significantly extended survival beyond either single agent alone (Fig. 8c,e). Digitoxin and MEK inhibitor thus also synergize to promote the survival of mice xenografted with human AML. This raises the possibility that disruption of ion homoeostasis may synergize with targeted agents to promote the regression of multiple cancers.

## Discussion

Cardiac glycosides exhibited limited single agent activity against primary human melanoma cells *in vivo* (Fig. 2a,d; Supplementary Fig. 3) despite the activity we observed in culture (Fig. 1a–c). However, cardiac glycosides combined with MEK inhibitor to inhibit NHE function (Fig. 4d), acidify the cytoplasm (Fig. 4a), increase mitochondrial $Ca^{2+}$ levels (Fig. 5c–f), reduce mitochondrial function (Fig. 6a,d), deplete ATP (Fig. 6h) and additively or synergistically increase cell death (Fig. 3a; summary in Fig. 7a). The net result was synergistically extended survival of mice xenografted with metastatic melanomas (Fig. 2d).

Either increased or decreased intracellular pH can increase intracellular $Ca^{2+}$ concentrations[44,45,55]. Our data suggest that treatment with digitoxin plus MEK inhibitor reduced intracellular pH, increasing intracellular $Ca^{2+}$ levels[44,45,55] and that $Ca^{2+}$ was sequestered to the endoplasmic reticulum and mitochondria[47,48]. Mitochondrial calcium overload and loss of membrane potential triggers mitochondrial transition pore opening and cell death[49,56]. Thus, the increase in mitochondrial $Ca^{2+}$ levels after treatment with digitoxin and MEK inhibitor may contribute to mitochondrial failure and cell death.

Digitoxin plus MEK inhibitor did not cause intracellular acidification, dysregulated mitochondrial $Ca^{2+}$ levels or cell death in normal human melanocytes or haematopoietic cells (Figs 3d–i, 4b and c, 5g–i). Cardiac glycosides plus MAPK pathway inhibitors may preferentially kill melanoma cells because melanoma cells are addicted to MAPK pathway signalling[4–7], exhibit elevated *ATP1A1* expression (Fig. 1e) and are more dependent on NHE function for maintenance of intracellular pH and survival[57]. Digitoxin plus MEK inhibitor likely also have other cellular effects that contribute to their ability to kill melanoma cells from some patients.

Current clinical data indicate that the MEK inhibitor trametinib extends progression-free survival of patients with BRAF mutant melanomas by 3.3 months beyond that observed with dacarbazine (1.5 months)[6] but not of patients with BRAF wild-type melanomas, irrespective of *NRAS* mutation status[58]. Our data raise the possibility that MEK inhibitor can be used to treat some BRAF wild-type melanomas if combined with a cardiac glycoside.

Our data demonstrate that disruption of ion gradients in cancer cells can synergize with MAPK pathway inhibitors to

**Figure 5 | Digitoxin and MEK inhibitor synergistically dysregulate mitochondrial $Ca^{2+}$ levels.** (**a**) Live-cell laser scanning confocal imaging of fluo-3 fluorescence in A375 melanoma cells to measure cytoplasmic $Ca^{2+}$. (**b**) Representative images of rhod-2 fluorescence in the mitochondria of cultured A375 cells before or 700 s after addition of 25 nM digoxin and 10 nM MEK inhibitor. Mitochondrial rhod-2 staining was confirmed by MitoTracker co-localization (Supplementary Fig. 7g). (**c–k**) Live-cell laser scanning confocal microscopy of rhod-2 fluorescence in the mitochondria of cultured A375 (**c**), M481 (**d**), M491 (**e**) or M214 (**f**) melanoma cells or normal human haematopoietic cells (**g**), two lines of primary human melanocytes (**h,i**) or cultured melanoma cells expressing vector (tRFP control) or NHE1 (**j,k**) to measure mitochondrial $Ca^{2+}$ levels. For each treatment 6–12 mitochondrial regions from each of 3–6 cells were continuously imaged for 30–37 min. The first arrow indicates the addition of 50 nM digitoxin and/or 10 nM MEK inhibitor. The second arrow indicates the addition of 10 μM FCCP to uncouple and dissipate mitochondrial membrane potential. Statistical significance was assessed by a two-way ANOVA followed by Dunnett's multiple comparisons test at 15 min (**c**) or 25 min (**d–k**). (**l,m**) Activated caspase-3+ cells in sections of M214 melanomas (**l**) and flow cytometric analysis of intracellular pH (**m**) in dissociated M214 xenografts from mice treated for four days with digitoxin plus MEK inhibitor and/or BAPTA-AM (a calcium chelator). Statistical significance was assessed by one-way ANOVA followed by Dunnett's multiple comparison's test. All data represent mean ± s.d. Each treatment compared with control (ns: not significant; $*P < 0.05$; $**P < 0.01$; $***P < 0.001$) or the combination compared with MEK inhibitor alone ($^{\#}P < 0.05$; $^{\#\#}P < 0.01$; $^{\#\#\#}P < 0.001$) or digitoxin alone ($^{\dagger}P < 0.05$; $^{\dagger\dagger}P < 0.01$; $^{\dagger\dagger\dagger}P < 0.001$).

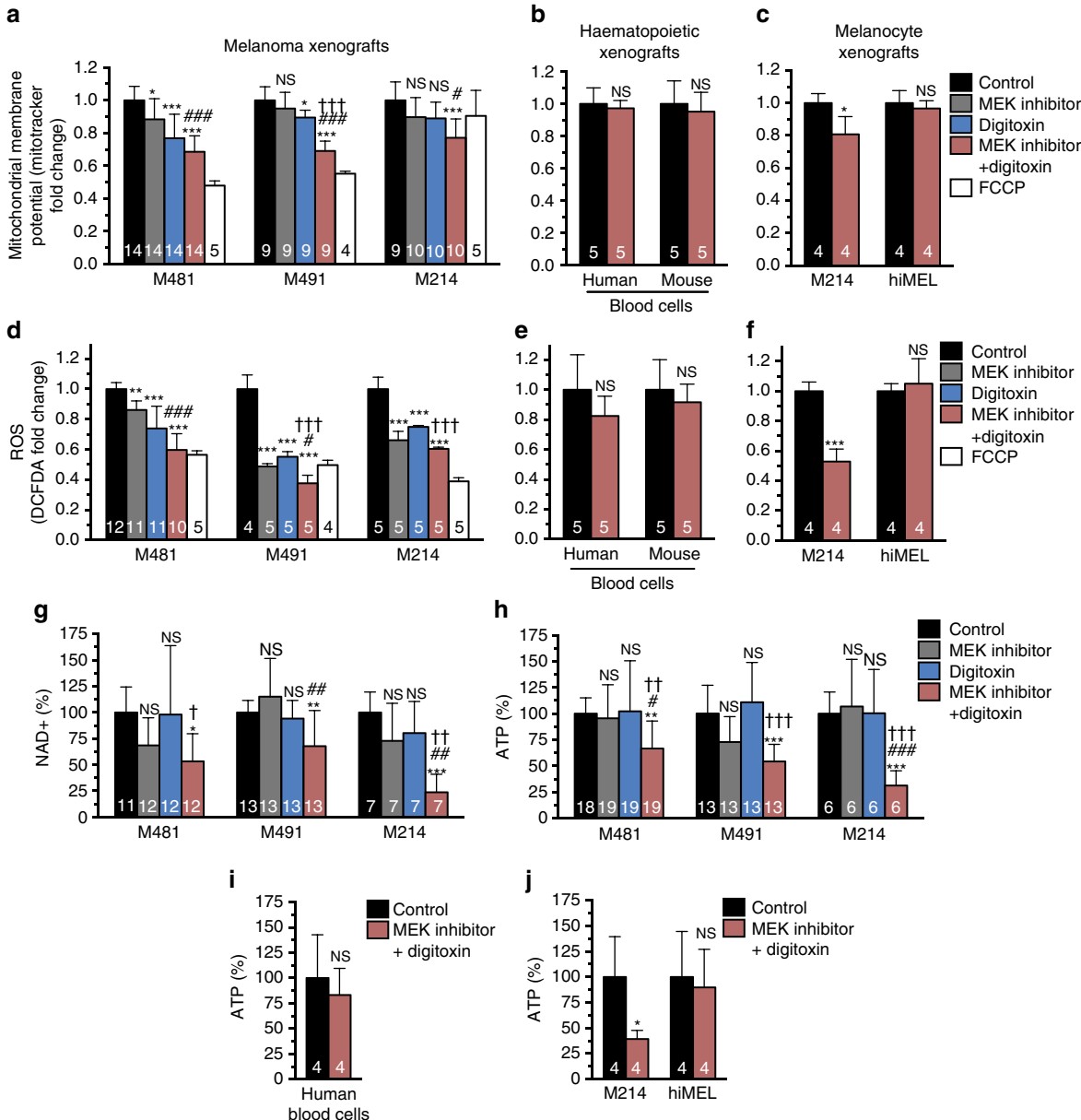

**Figure 6 | Digitoxin and MEK inhibitor additively or synergistically impair mitochondrial function and deplete ATP.** (**a–c**) Flow cytometric analysis of mitotracker deep red fluorescence. (**a**) Patient-derived xenografts were treated for four days *in vivo* with MEK inhibitor and/or digitoxin then acutely dissociated and live melanoma cells were analysed by flow cytometry (2–3 independent experiments per melanoma). (**b**) Mitochondrial membrane potential in human (HLA$^+$mCD45$^-$) or mouse (HLA$^-$mCD45$^+$) cells in the blood of NSG mice xenografted with hUCB and treated with digitoxin plus MEK inhibitor for 10 days. (**c**) Mitochondrial membrane potential of M214 melanoma cells and hiMEL immortalized melanocytes in gelatin sponges transplanted subcutaneously into opposing flanks of mice treated with digitoxin and/or MEK inhibitor for four days. (**d–f**) Flow cytometric analysis of relative DCFDA fluorescence to measure ROS levels in freshly dissociated melanoma cells (**d**, 1–3 independent experiments per melanoma), haematopoietic cells (**e**) or immortalized melanocytes (**f**) from mice treated with drugs for 4 days *in vivo*. (**g,h**) Relative NAD$^+$ (**g**) and ATP (**h**) levels in 150,000 live melanoma cells sorted from acutely dissociated tumours from 4-day treated mice (each melanoma was tested in 1–2 independent experiments). (**i**) Relative ATP levels in human haematopoietic cells sorted from the femurs of NSG mice xenografted with hUCB cells and treated with digitoxin plus MEK inhibitor for 4 days. (**j**) Relative ATP levels in M214 melanoma cells or immortalized melanocytes grown in gelatin sponges transplanted subcutaneously in opposing flanks of NSG mice and treated with digitoxin and/or MEK inhibitor for 4 days. All data represent mean ± s.d. In each panel the number of mice per treatment is written on the bars. In **a,d,g,h** statistical significance was assessed by one-way analysis of variance followed by Dunnett's multiple comparisons test or the Kruskal–Wallis test followed by Dunn's multiple comparisons test in cases of unequal variance among treatments. In **b,c,e,f,i,j** statistical significance was assessed by unpaired two-tailed *t*-tests. Statistical comparisons of each treatment versus control (NS, not significant; *$P < 0.05$; **$P < 0.01$; ***$P < 0.001$) or the combination compared with MEK inhibitor alone ($^\#P < 0.05$; $^{\#\#}P < 0.01$; $^{\#\#\#}P < 0.001$) or digitoxin alone ($^\dagger P < 0.05$; $^{\dagger\dagger}P < 0.01$; $^{\dagger\dagger\dagger}P < 0.001$).

promote tumour regression. Cancer cells may often be stressed with regard to their ability to maintain ion gradient homoeostasis such that cardiac glycosides may synergize with targeted agents in a range of cancers. Ion channels and transporters, including

*ATP1A1* (ref. 59), are recurrently mutated in certain cancers, particularly aldosterone producing adenomas[59–61]. These published data suggest that ion channel and transporter mutations can be drivers in certain contexts.

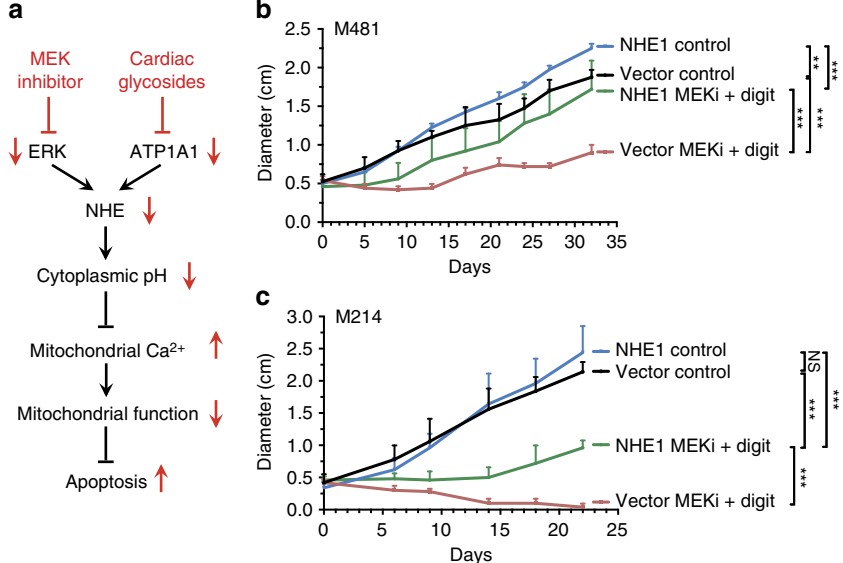

**Figure 7 | Digitoxin and MEK inhibitor inhibit tumour growth partly by inhibiting NHE function.** (**a**) Graphical summary of the mechanism by which digitoxin plus MEK inhibitor inhibit melanoma growth. (**b,c**) NSG mice with subcutaneous xenografts (M481 and M214) expressing vector (tRFP control) or NHE1 were treated daily with digitoxin and/or MEK inhibitor to examine the effect on tumour diameter ($n = 4$–$5$ mice per treatment per melanoma; each melanoma tested in an independent experiment). All data represent mean ± s.d. Statistical significance was assessed by two-way analysis of variance followed by Dunnett's multiple comparisons test (NS, not significant; $*P < 0.05$; $**P < 0.01$; $***P < 0.001$).

## Methods

**Tumour dissociation.** Melanoma specimens were obtained with informed consent from all patients according to protocols approved by the Institutional Review Board of the University of Michigan Medical School (IRBMED HUM00050754 and HUM00050085; see (ref. 10) for details). Tumours were dissociated in Kontes tubes with disposable pestles (VWR) followed by enzymatic digestion for 20 min with $200\,U\,ml^{-1}$ collagenase IV (Worthington), $5\,mM\,CaCl_2$ and $50\,U\,ml^{-1}$ DNase. To obtain a single-cell suspension, cells were filtered through a $40\,\mu m$ cell strainer.

**Primary melanoma cell cultures.** Freshly dissociated melanoma cells were cultured in ultralow attachment plates (Corning) with 'melanoma medium' containing 50% low-glucose Dulbecco's modified Eagle's medium, 30% Neurobasal (Invitrogen), 1% penicillin/streptomycin, 1% nonessential amino acids (Gibco), 50 mM 2-mercaptoethanol (Sigma), 1% N2 supplement, 2% B27 supplement (Gibco), recombinant human basic fibroblast growth factor ($20\,ng\,ml^{-1}$) and insulin-like growth factor 1 ($20\,ng\,ml^{-1}$) (R&D Systems).

**Viability assay.** Twenty-four hours before addition of drugs, $5 \times 10^3$ to $2 \times 10^4$ melanoma cells, hUCB cells (hUBC), melanocytes (hMEL), immortalized melanocytes (hiMEL) or A375 melanoma cells (ATCC) were seeded per well of white walled 96-well plates (Corning). Primary human melanocytes were either obtained from M. Soengas (Madrid; hMEL1), Life Technologies (hMEL2; lot number 1681215) or ATCC (hMEL3; lot number 60948598). Human immortalized melanocytes (hiMEL) were a gift from H. Widlund (immortalized by over-expression of hTERT, constitutively active CDK4 and a dominant negative fragment of p53)[23]. Cells were seeded either in melanoma medium (Fig. 1a,c and Supplementary Fig. 1b) or in dermal cell basal medium supplemented with adult melanocyte growth kit (ATCC; Fig. 1b,d and Supplementary Fig. 1a) or in DMEM supplemented with 2% fetal bovine serum (Sigma) and 1% penicillin/streptomycin (Fig. 1f and Supplementary Fig. 2a). Twenty-four hours later cardiac glycosides (gitoxin, digitoxin or digoxin; all from Sigma) were added to the cultures at the indicated doses. Cells were incubated for 72 h after addition of the compounds and assayed with Celltiter-Glo (Promega) according to the manufacturer's instructions using a plate reader FLUOstar Omega (BMG Labtech). For siRNA experiments, $10 \times 10^3$ A375 cells were transfected with 100 nM siRNAs (Dharmacon) against *ATP1A1*, *Ubiquitin B* (*UBB*, a positive control that is generally required by cells) or LON Peptidase N-Terminal Domain And Ring Finger 1 (*LONRF1*; a negative control that is generally not required by cancer cells) using RNAiMAX reagent (Life Technologies). After 4 days, cells were assayed with Celltiter-Glo as described above to determine viable cell numbers.

**Lentiviral transduction.** The human *ATP1A1*, *BCL2* and *NHE1* open reading frames were obtained from the Precision LentiORF collection (Dharmacon) in a bicistronic lentiviral expression construct that co-expressed turbo green fluorescent protein (for example, *pLOC-ATP1A1-IRES-tGFP*). As a control, turbo red

fluorescent protein (tRFP) was expressed (*pLOC-tRFP-IRES-tGFP*). The mouse *Atp1a1* open reading frame was PCR amplified from the mammalian gene collection (MGC:37763; gift from L. Lum) and cloned into pLOC control vector (*pLOC-tRFP-IRES-tGFP*) in place of tRFP. The primers used to amplify mouse *Atp1a1* (forward: 5′-TACCGAGCTCGGATCCtctagtctccagcaacagga-3′ and reverse: 5′-CGGTTCATTAGCTAGCagggcagtgggctagtagta-3′) contained homologous sequences to the pLOC vector (indicated by capital letters). Two lentiviral constructs carrying either dsRed2 and luciferase (*dsRed2-P2A-Luc*) or GFP and luciferase (*GFP-P2A-Luc*) were generated for bioluminescence imaging. dsRED2 and GFP were amplified using 3′ primers containing P2A self-cleaving peptide sequences. Firefly luciferase was PCR amplified with a 5′ primer containing overlapping P2A sequences. The resulting products were purified and cloned into the FUW lentivrial expression construct.

The primers that were used for generating these constructs were: dsRed2 forward, 5′-CGACTCTAGAGGATCCatggatagcactgagaacgtc-3′ (capital letters indicate homology to FUW backbone); dsRed2 reverse, 5′-TCCACGTCTCTCCAGCCTGCTTCAGCAGGCTGAAGTTAGTAGCTCCGCTTCCctggaacaggtggtggc-3′ (capital letters indicate P2A sequences); GFP forward, 5′-CGACTCTAGAGGATCCatggtgagcaagggcgagga-3′ (capital letters indicate homology to FUW backbone); GFP reverse, 5′-TCCACGTCTCTCCAGCCTGCTTCAGCAGGCTGAAGTTAGTAGCTCCGCTTCCcttgtacagctcgtccatgccg-3′ (capital letters indicate P2A sequences); luciferase forward, 5′-GCCTGCTGAAGCAGGCTGGAGACGTGGAGGAGAACCCTGGACCTGGATCCatggaagacgccaaaaacataaag-3′ (capital letters indicate P2A sequences) and luciferase reverse, 5′-GCTTGATATCGAATTCttacacggcgatctttccgc-3′ (capital letters indicate homology to FUW backbone). All constructs were generated using the In-Fusion HD cloning system (Clontech) and sequence verified.

For virus production, $0.9\,\mu g$ of the appropriate plasmid together with $1\,\mu g$ of helper plasmids ($0.4\,\mu g$ pMD2G and $0.6\,\mu g$ of psPAX2) were transfected to 293 T cells using polyjet (Signagen) according to the manufacturer's instructions. The resulting replication incompetent viral supernatants were collected at 48 h after transfection and filtered through a $0.45\,\mu m$ filter. 300,000 freshly dissociated melanoma cells were infected with viral supernatants supplemented with $10\,\mu g\,ml^{-1}$ polybrene (Sigma) for 4 h. Cells were then washed twice with staining medium (L15 medium containing bovine serum albumin ($1\,mg\,ml^{-1}$), 1% penicillin/streptomycin and 10 mM Hepes (pH 7.4)), and about 25,000 cells (a mixture of infected and non-infected cells) were suspended in staining medium with 25% high-protein Matrigel (product 354248; BD Biosciences) then injected subcutaneously into NSG mice. After growing to 1 to 2 cm in diameter, tumours were excised and dissociated into a single-cell suspension as described above. Dissociated melanoma cells were stained with directly conjugated antibodies against mouse CD45 (1:200, 30-F11-APC; eBiosciences), mouse CD31 (1:200, 390-APC; BioLegend) and mouse Ter119 (1:200, TER-119-APC; eBiosciences) for 20 min on ice to eliminate mouse haematopoietic and endothelial cells from sorts and analyses. Cells were washed with staining medium and re-suspended in 4′,6-diamidino-2-phenylindole (DAPI; $1\,\mu g\,ml^{-1}$; Sigma) to eliminate dead cells from sorts and analyses. GFP$^+$ CD45$^-$ CD31$^-$ TER119$^-$ DAPI$^-$ or

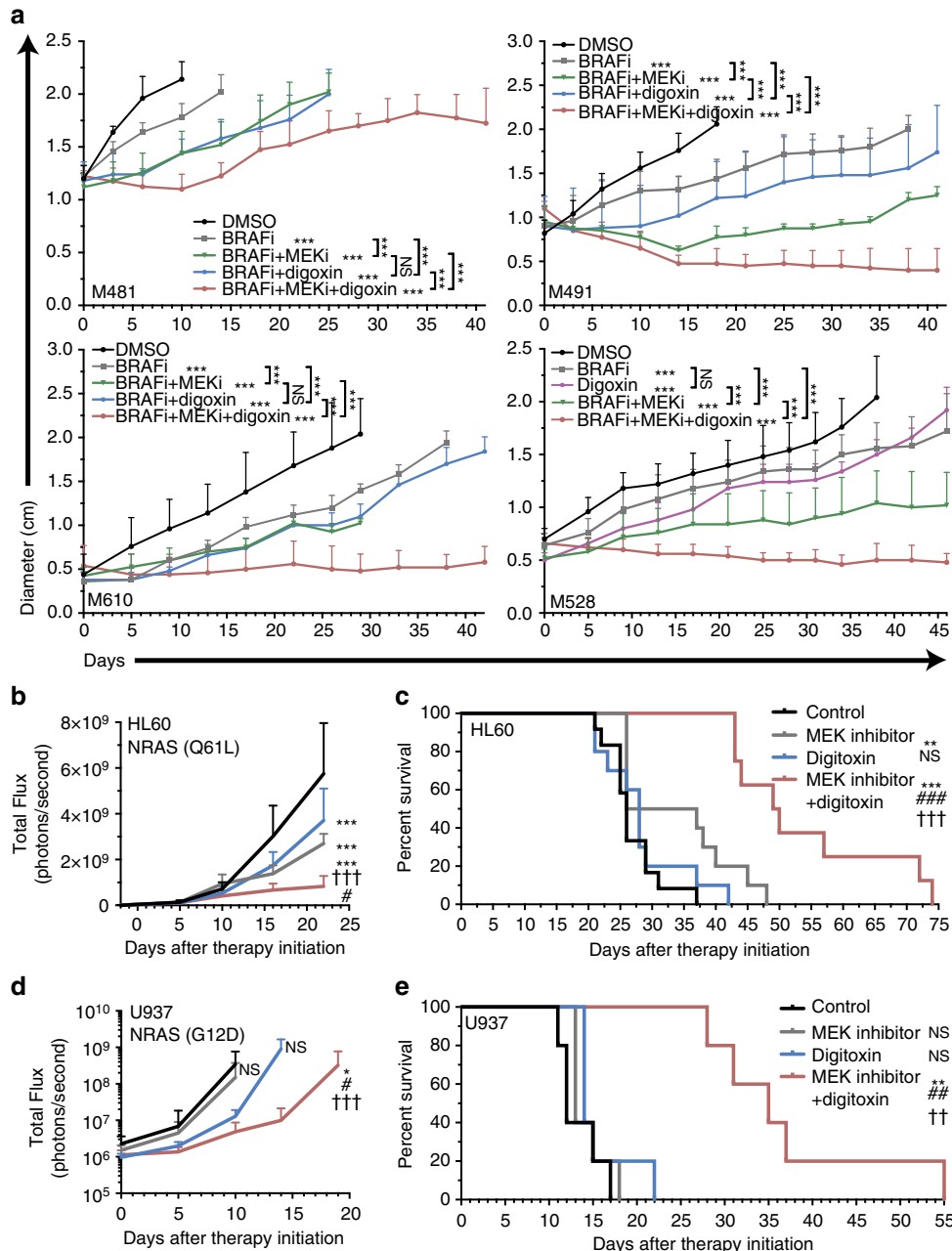

**Figure 8 | Combination therapy with digoxin also improves outcomes in mice xenografted with *BRAF^{V600E}* melanomas or *NRAS* driven human AML cells.** (**a**) NSG mice xenografted with four different BRAF mutant melanomas were treated with digoxin (10 mg kg$^{-1}$ body mass per day) and/or BRAF inhibitor (dabrafenib; 20 mg kg$^{-1}$ body mass per day) and/or MEK inhibitor (0.5 mg kg$^{-1}$ body mass per day) ($n = 4$–5 mice per treatment per melanoma; each melanoma was tested in an independent experiment). (**b**–**e**) Survival of NSG mice transplanted with HL60 (**b**,**c**) or U937 (**d**,**e**) human AML cells ($n = 5$–12 mice per treatment). (**b**,**d**) Disease burden was measured by bioluminescence imaging. Data in **a**,**b**,**d** represent mean ± s.d. Statistical significance was assessed by extra-sum-of-squares F tests (**a**,**d**) or by log-rank tests (**c**,**e**) followed by Bonferroni's multiple comparison's tests or a two-way analysis of variance followed by Dunnett's multiple comparisons test (**b**). Each treatment compared with control (NS: not significant; *$P < 0.05$; **$P < 0.01$; ***$P < 0.001$) or the combination compared with MEK inhibitor alone (#$P < 0.05$; ##$P < 0.01$;) or digoxin alone (†$P < 0.05$; ††$P < 0.01$; †††$P < 0.001$).

dsRed2$^+$CD45$^-$CD31$^-$TER119$^-$DAPI$^-$ virally infected live melanoma cells were sorted using a FACSAria (Becton Dickinson). The virally infected sorted cells were either injected into NSG mice for *in vivo* studies or added to cell culture for *in vitro* experiments.

**Xenograft assays.** All animal experiments were performed according to protocols approved by the Institutional Animal Care and Use Committees at the University of Texas Southwestern Medical Center (protocol 2011-0118). One million freshly dissociated melanoma cells were injected subcutaneously into the right flanks of NSG mice. When tumours become palpable, the mice were randomized to four different treatments. Mice were treated either with digoxin (Sigma, 0.5 mg kg$^{-1}$

body mass per day in 200 μl of 0.5% promethylcellulose, 0.2% Tween80 and 5% DMSO by oral gavage) or the MEK inhibitor trametinib (Selleckchem, 0.5 mg kg$^{-1}$ body mass per day in 200 μl of 0.5% promethylcellulose, 0.2% Tween80 and 5% DMSO by oral gavage) alone or in combination. Control mice received vehicle only (200 μl 0.5% promethylcellulose, 0.2% Tween80 and 5% DMSO per day, also by gavage). Experiments that used digoxin instead of digitoxin were performed with subcutaneous osmotic pumps (10 mg kg$^{-1}$ body mass per day; Alzet) due to the short half-life of digoxin in mice. The BRAF inhibitor vemurafenib was delivered in the chow, by mixing 417 mg of PLX4720 (Plexxikon) per kg of rodent diet (AIN-76 A; Research Diets). Control mice received AIN-76A chow without PLX4720. In experiments testing the triple combination of digitoxin plus BRAF and MEK inhibitors, the BRAF inhibitor dabrafenib (ChemieTek) was used instead of

vemurafinib. Dabrafinib was administered by oral gavage (20 mg kg$^{-1}$ body mass per day in 200 µl of 0.5% promethylcellulose, 0.2% Tween80 and 5% DMSO). Tumours were measured every 3–4 days with calipers. Experiments were terminated when any tumour in the cohort reached 2 cm in size then tumours were excised measured, weighed and photographed.

The effect of treatment with digitoxin and/or MEK inhibitor on the survival of mice with metastatic melanoma or human AML was assessed using cells that expressed GFP or DsRed as well as luciferase so that tumour burden could be measured by bioluminescence imaging. One hundred GFP$^+$CD45$^-$CD31$^-$TER119$^-$DAPI$^-$ or dsRed2$^+$CD45$^-$CD31$^-$TER119$^-$DAPI$^-$ melanoma cells in 25% matrigel were injected subcutaneously into the right flanks of NSG mice. Tumours were surgically excised when their largest diameter reached 2 cm. Mice were allowed to recover for a week and then randomized to receive daily treatment with trametinib and/or digitoxin as described above. $2 \times 10^6$ HL60 (ATCC) or U937 AML cells (ATCC) were intravenously injected to NSG mice through the tail vein. U937 is listed in the ICLAC database of commonly misidentified cell lines. However, authentic stocks are known to exist and we obtained the U937 cell line from ATCC, which authenticated it by DNA short-tandem repeat analysis. Mycoplasma testing of cancer cell lines was not performed. Mice were randomized to receive daily treatment with trametinib and/or digitoxin 1–2 weeks after transplantation. Mice were imaged on day 0 (before the initiation of therapy) and every 1–3 weeks by bioluminescence imaging. Briefly, mice were injected intraperitoneally with 100 µl of phosphate-buffered saline containing D-luciferin monopotassium salt (40 mg ml$^{-1}$; Biosynth) and imaged on an IVIS Imaging System 200 Series (Caliper Life Sciences) with Living Image software. For survival analysis, mice were monitored daily for signs of distress and killed according to a standard body condition score. The presence of macroscopic metastatic melanomas at distant sites was assessed at necropsy.

The effect of digitoxin plus MEK inhibitor on immortalized melanocytes (hiMEL) in vivo was determined by subcutaneously implanting absorbable gelatin sponges (Gelfoam, Pfizer) bearing hiMEL cells into mice. Forty-eight hours before transplantation, 100,000 melanoma cells (M214) or hiMEL cells in 150 µl of dermal cell basal medium supplemented with an adult melanocyte growth kit (ATCC) were seeded into 5x5 mm gelatin sponges. Each mouse was subcutaneously surgically transplanted with two sponges in opposite flanks, one containing M214 cells and the other containing hiMEL cells. Three days after transplantation mice were randomized to receive digitoxin plus MEK inhibitor or DMSO (control) for four days. Ninety-minutes after the last drug treatment, the sponges were surgically removed and washed in Ca$^{2+}$ and Mg$^{2+}$ free Hank's buffered salt solution (HBSS-free; Gibco). The cells were recovered from the sponges by incubating in 200 µl of 0.25% trypsin (HyClone) for 5 min at 37 °C. After quenching trypsin by adding 500 µl DMEM supplemented with 10% fetal bovine serum, cells were washed with HBSS-free and counted using a hemocytometer. Dissociated cells were stained with directly conjugated antibodies against mouse CD45, mouse CD31, mouse Ter119, as described above and human HLA (1:10, 555552, Becton Dickinson) for 20 min on ice to determine the frequency and absolute number of live (DAPI$^-$) human (HLA$^+$CD45$^-$CD31$^-$Ter119$^-$) cells in the sponges.

Experiments to determine the effect digitoxin plus MEK inhibitor on human haematopoietic cells in vivo were performed by transplanting 10$^7$ hUCB cells into the left femurs of sublethally (145 rads) irradiated NSG mice (female, 6–8 weeks). Cord blood was obtained through a human tissue acquisition and biorepository program maintained by the Department of Obstetrics and Gynecology through a protocol approved by the UT Southwestern Medical Center Institutional Review Board. Three to four weeks later, after the mice had engrafted with human haematopoiesis, mice were randomized to receive digitoxin plus MEK inhibitor or DMSO (control) for 4–10 days. After drug treatment, blood was collected from live mice by tail vein bleeding with the exception of Fig. 3h where cardiac puncture was used after the mice were killed. Bone marrow cells were isolated by flushing both femurs with HBSS-free supplemented with 2% heat-inactivated bovine serum and counted using a Vi-CELL cell viability analyser (Beckman Coulter). Antibody staining was performed as described above to flow cytometrically analyse the frequency and number of human haematopoietic cells in the bone marrow and blood.

**Testing whether the effects of digitoxin were on-target.** We addressed this question by testing whether over-expression of mouse *Atp1a1* could rescue the effects of digitoxin on the viability and membrane polarity of melanoma cells in culture. Melanoma cells expressing *mAtp1a1* or *hATP1A* as a result of lentiviral transduction were isolated either from primary NSG recipients (M481, M491 and M214) or from culture (A375 cells). To assess plasma membrane polarization, 100,000 melanoma cells in DMEM supplemented with 2% fetal bovine serum (Sigma) and 1% penicillin/streptomycin were treated with the indicated concentrations of digitoxin or DMSO (control) for 24 h. Cells were then loaded with 1 µM bis(1,3-diethylthiobarbituric acid)trimethine oxonol (DiSBAC$_2$(3), Life Technologies)[26] in HBSS-free for 30 min at room temperature. Cells were washed and re-suspended in HBSS-free with DAPI and fluorescence was measured with a FACSCanto II flow-cytometer (BD Biosciences). Data were analysed by FlowJo (Tree Star) software. To assess viability, 5,000 melanoma cells were seeded into wells of 96-well plates in quadruplicate and treated with the indicated concentrations of digitoxin or DMSO for 72 h. Viability was measured with Celltiter-Glo as described above.

**TUNEL and caspase assays.** Melanoma xenografts were excised and fixed in 10% neutral-buffered formalin, paraffin-embedded and sectioned. Terminal deoxynucleotidyltransferase-mediated UTP end label (TUNEL) staining for dead cells was done according to the manufacturer's instructions (Promega DeadEnd Fluorometric TUNEL System). Nuclei were co-stained with propidium iodide and then imaged and counted using an Olympus immunofluorescence microscope. To detect activated caspase-3 expression by cells in sections, formalin fixed paraffin-embedded slides were de-waxed with xylene and rehydrated with sequential washes with decreasing concentrations of ethanol. Antigen retrieval was performed in 10 mM sodium citrate buffer (pH 6.0), with heating by microwave for 20 min. Slides were treated with 3% hydrogen peroxide in methanol for 10 min to inhibit endogenous peroxidase activity then blocked with 5% goat serum and avidin/biotin blocking (Vector Laboratories; product SP-2001). They were then incubated with anti-cleaved caspase-3 antibody (Cell Signaling Technology; 5A1E) overnight at 4 °C followed by washing and incubation in biotinylated anti-rabbit secondary antibody (Vector Laboratories) for 30 min at room temperature. Staining was visualized using horseradish peroxidase (Elite ABC kit (product PK-6100) with diaminobenzidine substrate (product SK4105) both from Vector Laboratories). To detect caspase-3/7 activity in apoptotic cells a fluorogenic caspase-3/7 substrate was added to live cells and analysed by flow cytometry according to the manufacturer's instructions (CellEvent Caspase-3/7 Green Flow Cytometry Kit, Invitrogen).

**Western blot analysis.** Melanoma xenografts were excised and quickly snap frozen. Tissue lysates were prepared in Kontes tubes with disposable pestles using RIPA Buffer (Cell Signaling Technology) supplemented with phenylmethylsulphonyl fluoride (Sigma), and protease and phosphatase inhibitor cocktail (Roche). The bicinchoninic acid protein assay (Thermo) was used to quantify protein concentrations. Equal amounts of protein (10–20 µg) were separated on polyacrylamide gels and transferred to polyvinylidene difluoride membranes (BioRad). Membranes were blocked for 1 h at room temperature with 5% milk in TBS supplemented with 0.1% Tween20 (TBST) and subsequently incubated with primary antibodies overnight at 4 °C. After incubating with horseradish peroxidase conjugated secondary antibodies (Cell Signaling Technology), membranes were developed using SuperSignal West Pico or Femto chemiluminescence reagents (Thermo). Antibodies against pERK (p44/42 MAPK; Thr202/Tyr204; D13.14.4E) and ERK (p44/42 MAPK;137F5) were obtained from Cell Signaling Technologies. Antibodies against ATPA1A1 (C464.6), BCL2 (C-2) and Tubulin (TU-02) were obtained from Santa Cruz and used for western blot analyses. Uncropped scans of the main blots appear in Supplementary Fig. 8.

**Immunohistochemistry.** Human tissue sections corresponding to normal skin, benign nevi, primary melanomas and metastatic melanomas were stained with hematoxylin and eosin as well as with an ATP1A1 antibody (1:100; C464.6) following standard protocols. These slides were obtained as extra cuts from historical samples (originally obtained during routine care) that were available in our Dermatology department. We had no interaction with these patients and all samples were de-identified. On the basis of the UTSW guidelines (http://www.ut-southwestern.edu/research/research-administration/irb/about/index.html) and the US Department of Health and Human Services, this does not constitute human subjects research (HHS Regulation 45 CFR 46.102) and as such informed consent from the patients was not required.

**Biochemical Assays.** Melanomas were generally dissociated enzymatically, except when performing flow cytometric assays on tumours that were treated with drugs in vivo. In those cases, we obtained single-cell suspensions as rapidly as possible by performing only mechanical dissociation. In these cases, melanoma xenografts were surgically excised and placed in Kontes tubes containing 700 µl of staining medium. Melanomas were dissociated in staining medium using disposable pestles and single-cell suspensions were obtained by passing the cells through a 40-µm cell strainer. Stained haematopoietic cells or dissociated tumour cells (100,000–500,000) from mice treated with DMSO (control), MEK inhibitor alone, digitoxin alone or digitoxin plus MEK inhibitor were loaded with dyes to assess membrane potential, intracellular pH, glucose uptake or ROS levels. The dyes that were used to assess these parameters were all obtained from Life Technologies. We stained the dissociated cells for 30 min at room temperature with 1 µM bis(1,3-dibutylbarbituric acid)trimethine oxonol (DiBAC$_4$(3)) in HBSS-free (Ca$^{2+}$ and Mg$^{2+}$ free) to assess plasma membrane potential[26] or with 10 µM 5-(and-6)-Carboxy-Seminaphthorhodaflouor-1 (Acetoxymethyl Ester) (SNARF1) in HBSS-free[62] to assess intracellular pH. We stained the dissociated cells for 15 min at 37 °C with 100 nM MitoTracker Deep Red FM to assess mitochondrial membrane potential ($\Delta\Psi_m$)[50] or with 5 µM 2′–7′-dichlorofluorescein diacetate (DCFDA) in HBSS-free to assess ROS levels[52]. After staining, the cells were washed and analysed either with a FACSAria or a FACSCanto using appropriate filter sets and excitations for each indicator. Data gated on live melanoma cells were analysed using FlowJo software. Controls included 150 mM KCl to depolarize the plasma membrane and 10 µM carbonyl cyanide-p-trifluoromethoxyphenylhydrazone (FCCP, Santa Cruz) to dissipate $\Delta\Psi_m$. To determine actual intracellular pH we generated standard curves with all treatments and melanomas by incubating cells with pH 5.5, pH 6.5 or pH 7.5 buffers in the presence of 10 µM valinomycin and nigercin (ionophores

that allowed the cytoplasm to equilibrate with extracellular pH; Life Technologies). SNARF1 fluorescence obtained from cells was then converted to pH values using these standard curves. To assess NHE activity five million melanoma cells were stained with SNARF1 as described above and suspended in bicarbonate free DMEM with 5 mM glucose and 25 mM HEPES (pH 7.4). After recording the baseline fluorescence for 60 s, the run was paused and intracellular acidification was induced by adding 50 μl of sodium propionate from a 1 mM stock solution prepared in the same medium. Fluorescence was recorded for an additional 300 s to monitor recovery from intracellular acidification. Changes in intracellular pH during the experiment were analysed using the kinetics function of FlowJo (Tree Star) software and the rate of recovery was expressed as $\Delta$pH s$^{-1}$. Amiloride (20 mg kg$^{-1}$ body mass per day; Sigma) and BAPTA-AM (6 mg kg$^{-1}$ body mass per day; Sigma) were intraperitoneally injected in saline.

**Live-cell imaging of mitochondrial Ca$^{2+}$ levels.** Melanoma cells were cultured in glass-bottom dishes (No. 1.0, MatTek Corp.) with DMEM supplemented with 10% fetal bovine serum (M481, M491 and A375) or in dermal cell basal medium supplemented with adult melanocyte growth kit (for M214 melanoma cells) or in RPMI supplemented with 10% fetal bovine serum (for hUCB) for at least 24 h to promote cell attachment and spreading. In the case of hUCB all of the attached cells were confirmed to be haematopoietic by staining with anti-hCD45. Cells were then washed with HBSS (containing Ca$^{2+}$ and Mg$^{2+}$) and loaded with 5 μM Rhod-2 (ref. 49) (AM ester, Life Technologies) for 20 min at room temperature. Cells were washed, then HBSS was added to each dish and each dish was placed into the imaging chamber of an inverted Zeiss LSM 780 laser scanning confocal microscope. Images were acquired every 10 s. At the time indicated in the figures, DMSO, cardiac glycosides and/or MEK inhibitor were added to the dish. After imaging for 18–30 min, the cells were treated with 10 μM FCCP to depolarize and deplete Ca$^{2+}$ inside mitochondria. Mitochondrial localization of Rhod-2 was also confirmed on cells by assessing co-localization with Mitotracker deep red (100 nM for 20 min at room temperature). To avoid signal originating from the nucleus we used ImageJ software (NIH) to draw regions of interest around mitochondrial localized Rhod-2 and to quantify fluorescence only in these regions. To assess cytoplasmic Ca$^{2+}$ levels, cells were loaded with 2 μM Fluo3 (ref. 46) (AM ester, Life Technologies) for 30 min at room temperature and processed as described above. After imaging for 22 min, the cells were treated with 5 μM thapsigargin to raise intracellular free Ca$^{2+}$.

**Measurement of ATP and NAD$^+$ levels.** 150,000 live human melanoma cells or 10,000 live human haematopoietic cells were isolated from xenografted mice by flow cytometry as described above then the cells were pelleted by centrifugation. The pellets were lysed in buffers from BioVision kits designed to quantitate NAD$^+$ or ATP levels. A colorimetric assay using an NAD$^+$/NADH cycling reaction was used to detect NAD$^+$ levels (NAD$^+$/NADH Quantitation Colorimetric Kit, BioVision)[53] according to the manufacturer's instructions. NADP and NADPH levels were measured using a luciferase-based assay using a similar cycling reaction (NADP/NADPH-Glo; Promega). ATP levels were quantitated using a luciferase-based assay in which luciferase activity depends on ATP from the lysed cells (Biovision)[54].

**Gene expression analysis.** Melanomas were surgically excised and homogenized in Trizol (Invitrogen), then frozen at $-80\,^{\circ}$C. After isolation of RNA from Trizol, total RNA from each sample was further purified using RNeasy micro plus kits (Qiagen). For microarray analysis, 1 μg of total RNA was biotin labelled using the Illumina Total Prep kit (Ambion) according to the manufacturer's instructions. The resulting cRNA was then analysed with the Illumina Human HT12v4.0 Expression Beadchip (Illumina) at the University of Texas Southwestern Medical Center (UTSW) Simmons Cancer Center Genomics Core. The arrays were scanned using an Illumina Beadstation 500 BeadArray reader. BeadStudio (Illumina) was used for data analysis. After quantile normalization, signal intensities for ATP1A1, ATP1A2, ATP1A3 and ATP1A4 were compared between melanomas and melanocytes.

***In vivo* isotope tracing.** Mice were fasted for 16 h before intraperitoneal injection of D[U-$^{13}$C] glucose (Cambridge Isotopes; 2 g kg$^{-1}$ body mass) and analysed 30 min later. Tumours were surgically excised, immediately placed in ice-cold 50% methanol in watch and homogenized. Metabolites were extracted by three cycles of freezing in liquid nitrogen and thawing in a 37 $^{\circ}$C water bath. Macromolecules and debris were removed by centrifugation and the supernatants containing soluble metabolites were dried and derivatized for 30 min at 42 $^{\circ}$C in 100 μl of a tri-methylsilyl donor (Tri-Sil, Thermo). Metabolites were analysed using an Agilent 6970 gas chromatograph networked to an Agilent 5973 mass selective detector. $^{13}$C enrichment analysis was performed as previously described[63].

**Statistical methods.** In most cases, multiple melanomas from different patients were tested in multiple independent experiments performed on different days. We always indicated variation using standard deviation. For analysis of statistical significance, we first tested whether there was homogeneity of variation across treatments (as required for ANOVA) using Levene's test. In cases where the variation significantly differed among treatments, the data were log$_2$-transformed. If the data contained zero values, ½ of the smallest non-zero value was added to all measurements before log$_2$ transformation. If the data contained negative values, all measurements were log-modulus transformed (L(x) = sign(x) * log$_2$(|x| + 1)). In rare cases when transformed data continued to exhibit significantly different variation among treatments, we used a non-parametric Kruskal–Wallis test to assess the significance of differences among treatments. When variation did not significantly differ among treatments, two-tailed Student's t-tests were used to test the significance of differences between two treatments. When more than two treatments were compared, a one-way ANOVA followed by Dunnett's multiple comparisons tests were performed. A two-way ANOVA followed by Dunnett's multiple comparisons tests were used in cases where more than two groups were compared with repeated measures. In cases where the durations of the repeated measures were not the same for all groups (Fig. 8a,d, Supplementary Figs 4c and 6) the data were fitted using a non-linear regression (exponential growth equation Y = Y0*exp(k*X)) and groups were compared using extra-sum-of-squares F tests followed by Bonferroni's multiple comparisons tests. The log-rank test was used to assess the significance of differences in survival curves (Figs 2d and 8c,e and Supplementary Fig. 5b).

All in vivo experiments were randomized. No blinding was used in any experiment. In all in vivo experiments 3–8 week old female or male NSG mice were used. For short-term assays and xenograft assays we injected 3–5 mice per treatment. For long-term survival assays we injected 7–10 mice per treatment to account for non-melanoma related deaths (NSG mice are susceptible to death from opportunistic infections). There were only two experiments where mice were excluded. In Fig. 2a testing the effect of digitoxin and/or MEK inhibitor on the growth of xenografted melanomas, 0–2 mice per melanoma were found dead due to an opportunistic bacterial infection before termination of the experiment and were excluded from the reported results. In Fig. 2d and Supplementary Fig. 4b testing the effect of digitoxin and/or MEK inhibitor on the survival of mice with metastatic melanoma, 22 of 108 mice died before the termination of the experiment for reasons that did not appear to be melanoma-related—mainly opportunistic bacterial infections. These mice did not have detectable metastatic disease and were excluded from the survival analysis. In the same experiments, 9 mice developed re-growth at the site of the primary tumour after the surgery. These mice were included in the survival analysis (Fig. 2d) but excluded from total flux quantification by bioluminescence (BLI) imaging (Supplementary Fig. 4c,d) since the BLI imaging was designed to quantify metastatic disease burden and the subcutaneous tumour re-growth made it impossible to assess metastatic disease burden by BLI.

**Data Availability.** The authors declare that data supporting the findings of this study are available within the article and its Supplementary Information files or from the corresponding author on request. The microarray data cited in Fig. 1 and Supplementary Fig. 1 have been deposited in the NCBI GEO database under the accession code GSE83583.

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

## Acknowledgements

S.J.M. is a Howard Hughes Medical Institute (HHMI) Investigator, the Mary McDermott Cook Chair in Pediatric Genetics, the director of the Hamon Laboratory for Stem Cells and Cancer, and a Cancer Prevention and Research Institute of Texas Scholar. In addition to support from HHMI and CPRIT to S.J.M., this work was also supported by the National Institutes of Health (CA157996) to R.J.D. M.W. was supported by a British Society for Haematology Scientific Fellowship. We thank the Department of Obstetrics and Gynecology for providing umbilical cord blood (a service supported, in part, through NIH P01HD11149) and to the Michigan Biobank (supported in part by the Lewis and Lillian Becker and Howard Cooper Funds). We thank N. Meireles and A. Durham for tissue procurement and clinical data management; J. Peyer and S. Mann for technical support; K. Correll and M. Gross for mouse colony management; N. Loof and the Moody Foundation Flow Cytometry Facility; Z. Hu for digitoxin pharmacokinetics; K. Phelps for live-cell imaging and J. Shelton for histology.

## Author contributions

M.S. and E.Q. first noticed an effect of cardiac glycosides on cultured melanoma cells, established the initial tumour xenografts and performed the microarray analysis. U.E. confirmed the effect and detected synergy with MAPK pathway inhibitors. U.E. performed all the experiments with assistance from V.R. Z.Z. analysed RNA sequencing data and performed statistical tests. S.W.Y. helped to perform experiments in culture. M.W. performed the experiment to examine activity against leukaemia cells. J.G.G and T.V. performed the immunohistochemical analysis of ATP1A1 expression. T.M.J. provided melanoma specimens and associated clinical data. U.E. and S.J.M participated in the

design and interpretation of all experiments. R.J.D. participated in the design and interpretation of experiments exploring metabolic mechanisms. U.E. and S.J.M wrote the manuscript.

## Additional information

**Competing financial interests:** The authors declare no competing financial interests.

**How to cite this article**: Eskiocak, U. *et al.* Synergistic effects of ion transporter and MAP kinase pathway inhibitors in melanoma. *Nat. Commun.* 7:12336 doi: 10.1038/ncomms12336 (2016).

