## [Peer Review File · Nature Communications]

Reviewers' comments:

Reviewer #1 (cancer and mitochondrial function)(Remarks to the Author):

This is an excellent paper that demonstrates the utilization of cardiac glycosides as potential therapy for human melanoma. The observation that MEK or BRAF inhibition synergistically combines with digitoxin to induce cell death is a novel observation. The in vivo data is very compelling and the implications of the paper have direct clinical relevance. Biologically the paper also suggests that mitochondrial calcium overloading might be a novel mechanism to kill cancer cells.

I have one suggestion that the authors might want to consider to bolster their conclusions.

The authors suggest calcium overload as a mechanism of toxicity. I would suggest they test whether RU360, a calcium uniporter inhibitor, prevents the cell death.

Reviewer #2 (AML (in vivo, MEK/BRAF inhibitors))(Remarks to the Author):

Synergistic effects of ion transporter and MAP kinase pathway inhibitors in melanoma.

thanks for the opportunity to review this interesting manuscript. The authors have primarily used patient derived xenograft samples and in vivo treatment to demonstrate synergy between MEK inhibitors and digitalis based treatments to block NaK ATP pumps. The pump ATP1A1 is increased in melanoma cells and correlates with in vitro sensitivity to digitoxin. Digoxin, more commonly used clinically, was also tested in functional assays, at least initially. PDX experiments show mild to no phenotype with either agent alone, but striking efficacy in combination, including reduction in metastatic disease. This is reversed by expression of the mouse *Atp1a1* gene that is resistant to digitoxin. Mechanistically, the authors describe reduction in intracellular pH, inhibition of NHE activity, blocking of Ca^{++} pumps and accumulation of intracellular Ca^{++} leading to caspase release and mitochondrial depolarisation. The experiments are validated in vivo.

Overall this is a strong and comprehensive demonstration of synergy between 2 compounds with independent and non-overlapping function. The paper has direct clinical relevance and could (should) be used as a rationale to translate into clinical trials (and is NCT02138292). The data are clear, well presented and have appropriate statistics used throughout.

Minor comments only:

WB figure 1F is not convincing, repeat please with more samples from melanoma and normal controls (e.g. also include hiMEL and some additional melanoma samples).

In Supp Fig 5b, m*Atp1a1* only partially rescues the combination treatment phenotype. Is this therapeutic effect because of the MEKi part of the combination?

I find the data in Fig 8b-d a bit confusing. The rationale to specifically choose AML is not clear, and the authors now use cell lines (contrast with the PDX models used in the paper throughout). No mechanistic validation is provided for the AML experiments, which is in contrast to the detailed and elegant work in melanoma.

I would like to see the HIF1a data included in the manuscript as an important negative control, both at current dose and at higher dose.

Reviewer #3 (cardiac glycosides/ NHE)(Remarks to the Author):

General Comments:

This article elaborates on the effects and the potential underlying mechanisms that a combinational use of cardiac glycosides and MEK inhibitor, despite both had been previously investigated individually, to inhibit growth of melanoma cells in vitro and in vivo (xenografted mice) using primary human melanoma cells derived from human patients. The authors first demonstrated that cardiac glycosides preferentially inhibited melanoma cell growth than some normal human cells in vitro (cell culture), which might be related with a high expression level of the ATP1A1 Na⁺/K⁺ pump in melanoma cells. Although no correlation between the ATP1A1 mRNA levels and the IC50 of digitoxin was observed, overexpression of the cardiac glycosides-insensitive mouse ATP1A1 but not the sensitive human ATP1A1 can block the effects of digitoxin on viability and plasmamembrane depolarization in cultured melanoma cells. Then, with their developed xenograft assay, it was clearly shown that a combination of the digitoxin and MEK inhibitor trametinib was significantly more effective, 4 additively and 5-6 synergistically, than either of the single agent alone in 8 of 11 melanoma xenografts derived from patients. The author then investigated the mechanisms to look for causes of the synergistic effects of these two agents. The authors studied the effects of these two types of agents on cell plasma membrane depolarization, ERK phosphorylation in MARK pathway, glucose uptake and metabolism, cell size, intracellular pH, mitochondrial Ca²⁺ level, and mitochondrial function including mitochondrial membrane potentials, ROS level, NADP⁺ and ATP levels. The authors proposed that these two agents synergistically inhibited the Na⁺/H⁺ exchanger (NHE) activity, which led to decrease in intracellular pH, elevation in mitochondrial Ca²⁺ and thus mitochondrial function.

This manuscript involves a large amount of assays and data which appear to be carefully done in most experiments. It is clearly shown that a combinational use of these two agents was more effective in killing melanoma cells, lowering intracellular pH, inhibiting NHE activity, increasing mitochondrial Ca²⁺, and decreasing mitochondrial transmembrane potential, ROS, NAD⁺ and ATP levels. To this reviewer, the main concern of this manuscript is the proposed pivot role of NHE. Although it is safe to draw from the data (Fig. 4d) that digitoxin and MEK inhibitor alone had no effect on NHE activity but together showed strong synergetic effect particularly in M214 cells, the two pieces of evidence to suggest an involvement of NHE1 in intracellular pH and cell death are both weak. The use of a non-specific blocker amiloride to block NHE activity will also block other a family of epithelial Na⁺ channels and other Na⁺ dependent transporter and additionally, the effect was smaller than digitoxin plus MEK inhibitor. Secondly, the use of overexpression to rescue only showed very limited success. Additionally, the results from NHE activity assay seems to be inconsistent with those in intracellular pH assays, which seemed to be largely an additive effect of those two agents although the large error bars make it hard to say additive or synergetic. Inconsistence exists for M481 in digitoxin-induced effects on intracellular pH (Fig 4a) and NHE activity (Fig 4b). The downstream effects on mitochondrial function were also not included in these tests with amiloride and NHE overexpression. It will be more convincing if digitoxin plus MEK lost their most effects in cell death and mitochondrial function upon a knockdown of NHE1 expression. Therefore, the underlying mechanisms, which were probably more complicated, remained not well established to this reviewer.

Specific Comments:

1) additive vs synergetic-- they are clearly different terms; however due to errors/variations presented in the experimental data, sometimes it is hard to definitely to call an additive or a synergetic effect of the two agents. For example, in Fig4a, by eye it seems to be an additive effect but statistical analysis may show a synergetic effect which essentially cannot rule out an additive effect because of the large error issue. In Fig2a, the melanoma cells derived from different patients, as the authors stated, some seemed to be additive and some to be synergetic. The manuscript including the title seems to have an emphasis on synergetic effects. It might be necessary to carefully reexamine the additive and synergetic effects throughout the manuscript, or maybe more appropriate to deemphasize the synergetic effects.

Page 10, last sentence of first paragraph: cannot say "does not have additive" because it is kind of additive effect ($A \& B = A + B$ although A or $B = 0$).

2) The last sentence in the abstract "can synergize with targeted agents" seems to be unclear for the exact meanings.

3) It is impressive that a large number of different melanoma cells were used in the initial in vitro (from 15 patients) and in vivo (from 11 patients) cell growth assays. However, in the rest of the manuscript, only M481, M491 and M214 were selectively used. It will be better to explain somewhere why these 3 derived melanoma cells were preferentially used.

4) Fig 1f, not mentioned in the figure legend.

5) Fig 2a, It is a little odd that the M214's data only includes up to 9 days while all others show data extended to 25 days or more. Apparently it is not short of data. In the supplementary Fig 6, similar data of M214 does cover more than 25 days.

6) Fig 5e, it might be necessary to explain why FCCP had little effect on the MEK inhibitor + digitoxin -induced increase in mitochondrial Ca^{2+} . Additionally, because of the large value (~ 2000) of time in second, a label in minute might be better for the x-axis in Fig 5a, c-k.

7) Page 8, last paragraph. It was reported that apoptosis, autophagy and necrosis are involved in melanoma cells death induced by MEK inhibitor. Caspase-3 activation might be not the major cause for melanoma cell apoptosis. It will be helpful to explain why it is chosen as marker of cell death here.

8) page 9, first line "data not shown". It is better either not to mention the data or actually show the data if mentioned.

9) Page 10, last sentence of 2nd paragraph is not clear. The point should be there is no good correlation between the digitoxin plus MEK inhibitor -induced cell membrane depolarization, which also happened in non-melanoma cells, and the preferential toxicity of digitoxin plus MEK inhibitor toward melanoma cells.

10) page 11 and Fig 4a. The connection between intracellular pH and digitoxin plus MEK inhibitor-induced cell death is weak. The initial pH before treatment (\sim pH 7.3, 8.0, 7.0) and final pH after treatment (\sim pH 6.6, 7.2, 6.7) are very different among the used 3 cells, M481, M491 and M214. The difference is also very small before and after treatment in the M214 cells.

11) page 13. "Intracellular acidification increases intracellular Ca^{2+} levels by inhibiting various Ca^{2+} channels". It is better to correctly use the term "channel" and transporter". The former passively moves Ca^{2+} across the Ca^{2+} gradient, thus will decrease the intracellular Ca^{2+} after inhibition. The authors may mean transporters here.

The evidence for connection between intracellular pH decrease and mitochondrial Ca^{2+} elevation seems to be not strong here.

12) page 14. The use of BAPTA-AM is problematic and the data is hard to interpret because of its non-specific effects on all cellular Ca^{2+} .

Reviewer #4 (Melanoma (in vivo, therapy))(Remarks to the Author):

The manuscript by Eskiocak U et al. investigated the efficacy of the combination of cardiac

glycosides and MAPK inhibitors in treating melanoma in vitro and in vivo. The most difficult hurdle in the context of BRAF-targeted therapies is to overcome the acquisition of therapy resistance to MAPK inhibitors. The authors did not investigate the combination therapy presented in this manuscript toward that end.

Major comments are:

1. The reviewer is not convinced by the rationale of investigating the efficacy of cardiac glycosides in malignant melanoma, which has not been fully addressed.
2. Please strengthen data shown in Fig 1e by analyzing gene expression microarray data included in GSE46517 and TCGA SKCM. Break down samples into skin melanocytes, nevi, primary melanoma and metastatic melanoma (GSE46517) or primary melanoma and metastatic melanoma (TCGA SKCM).
3. Please strengthen data shown in Fig 1e and 1f by performing IHC staining of ATP1A1 in normal skin, nevi, primary melanoma and metastatic melanoma.
4. Fig 1g - ATP1A1 was expressed at a higher level in melanomas compared to that in melanocytes. Why not over expressing hATP1A1 in hMEL1 and hMEL2 and then treating them with Digitoxin?
5. It is absolutely critical that the authors perform the knock-down experiment in M481, M491 and M214 to demonstrate if siATP1A1 or shATP1A1 can recapitulate the efficacy of Digitoxin.
6. Fig 1g and 1h - the comparison made in this panel did not make sense. For example, the authors should have compared the control cells to cells over-expressing mAtp1a1 or hATP1A1 when treated with DMSO or Digitoxin at 25 or 50nM.
7. Fig 2a - please replace "Diameter (cm)" with "Tumor Volumes (mm³)". Basically, Digitoxin as a single agent did not work in most cases, which is not in line with in vitro data shown in Fig 1.
8. Fig 3d, 3f, 3g, 3h, 3i - MEKi or Digitoxin alone were missing. This shall also apply to the remaining figures when data of the single inhibitor control be included.
9. Fig 3j - data basically showed that the decrease in pERK when cells were treated with the combo was due to the treatment of MEKi, which did not support the additive effect or synergy between MEKi and Digitoxin. A more in-depth understanding of why the combo was working better than either single agent is clearly warranted in this study.
10. Fig 8 - Digoxin single agent was missing for M481, M491 and M610.
11. The authors used "Additively" and "Synergistically" throughout this manuscript. Be very careful about using these words if no scientific evidence are in place to support this claim. For the in vitro studies, the authors would have diluted either single agent in order to calculate Bliss values. The situation is more complicated for interpreting in vivo data. Please consider using "Significantly" or "Markedly".

Reviewer #1:

This is an excellent paper that demonstrates the utilization of cardiac glycosides as potential therapy for human melanoma. The observation that MEK or BRAF inhibition synergistically combines with digitoxin to induce cell death is a novel observation.

The in vivo data is very compelling and the implications of the paper have direct clinical relevance. Biologically the paper also suggests that mitochondrial calcium overloading might be a novel mechanism to kill cancer cells.

I have one suggestion that the authors might want to consider to bolster their conclusions.

The authors suggest calcium overload as a mechanism of toxicity.

I would suggest they test whether RU360, a calcium uniporter inhibitor, prevents the cell death.

RESPONSE: We thank the reviewer for this suggestion. Unfortunately, we have been unable to obtain RU360. We have attempted to order RU360 from two different sources and found that it is back ordered until January 2017 (EMD Millipore) or indefinitely (Santa Cruz). We have not been able to identify another RU360 supplier. Therefore, we are not able to perform this experiment. We also have not been able to find any examples of RU360 use in vivo in rodents, or any pharmacokinetic data that would guide the experiments. It isn't clear to us whether it would be tolerated in vivo or what concentration it should be used at. Resolving these questions would take months of control experiments, if we were able to obtain the drug.

Multiple independent lines of experiments in the current manuscript suggest that disruption of Ca^{2+} hemostasis contributes to the effects of digitoxin plus MEK inhibitor on melanoma cells. The increase in cell death induced by digitoxin plus MEK inhibitor was partially rescued by calcium chelation (Fig. 5l). However, calcium chelation did not rescue the decline in intracellular pH, suggesting that the increase in mitochondrial Ca^{2+} is downstream of the change in intracellular pH (Fig. 5m). Consistent with this, *NHE1* over-expression partially rescued the effects of digitoxin plus MEK inhibitor on intracellular pH (Fig. 4g), mitochondrial Ca^{2+} levels (Fig. 5j, k), cell death (Fig. 4h), and tumor growth (Fig. 7b, c).

Reviewer #2:

Synergistic effects of ion transporter and MAP kinase pathway inhibitors in melanoma.

thanks for the opportunity to review this interesting manuscript. The authors have primarily used patient derived xenograft samples and in vivo treatment to demonstrate synergy between MEK inhibitors and digitalis based treatments to block NaK ATP pumps. The pump ATP1A1 is increased in melanoma cells and correlates with in vitro sensitivity to digitoxin. Digoxin, more commonly used clinically, was also tested in functional assays, at least initially.

*PDX experiments show mild to no phenotype with either agent alone, but striking efficacy in combination, including reduction in metastatic disease. This is reversed by expression of the mouse *Atp1a1* gene that is resistant to digitoxin. Mechanistically, the authors describe reduction in intracellular pH, inhibition of NHE activity, blocking of Ca^{++} pumps and accumulation of intracellular Ca^{++} leading to caspase release and mitochondrial depolarisation. The experiments are validated in vivo.*

Overall this is a strong and comprehensive demonstration of synergy between 2 compounds with independent and non-overlapping function. The paper has direct clinical relevance and could (should) be used as a rationale to translate into clinical trials (and is NCT02138292). The data are clear, well presented and have appropriate statistics used throughout.

Minor comments only:

WB figure 1F is not convincing, repeat please with more samples from melanoma and normal controls (e.g. also include hiMEL and some additional melanoma samples).

RESPONSE: We have expanded the ATP1A1 western blot analysis to include more melanocytes and melanomas (see new Suppl. Fig. 1j). The expanded western analysis of normal melanocytes and melanoma samples shows that the immortalized melanocytes express ATP1A1 at higher levels than the normal melanocyte samples, and that 4 of 8 melanomas express ATP1A1 at higher levels than in any of the normal melanocyte samples but not higher than in the immortalized melanocytes (Suppl. Fig. 1j). Our microarray analysis also demonstrated that most melanomas express higher levels of *ATP1A1* as compared to normal melanocytes (Fig. 1e and Suppl. Fig. 1e-f).

We have further supplemented these data by adding a reanalysis of microarray data from two previously published studies from the NCBI GEO Database: GDS1375 (Clin Canc Res 11:7234) and GSE46517 (PLoS One 5:e10770). These data show that *ATP1A1* transcript levels tend to go up in benign nevi as compared to normal skin, though the differences were not statistically significant (Suppl. Fig. 1g and 1h). In contrast, *ATP1A1* transcript levels did significantly increase in malignant melanomas as compared to normal skin and benign nevi (Suppl. Fig. 1g and 1h). These independent results are consistent with our data in showing that *ATP1A1* transcript levels tend to increase in melanoma cells as compared to normal cells.

We have also added new data showing immunohistochemical staining for ATP1A1 in normal human skin, benign nevus, primary melanoma, and metastatic melanoma specimens (see new Suppl. Fig. 1i). ATP1A1 protein was limited to basal keratinocytes in normal human epidermis but expanded to include melanocytic nests in benign nevi (Suppl. Fig. 1i). In primary and metastatic melanomas, ATP1A1 staining was robust and nearly homogeneous among melanoma cells (Suppl. Fig. 1i). Since the western data from dissected tumors included variable amounts of melanoma cells and normal stromal cells, these IHC data more definitively demonstrate that ATP1A1 expression tends to increase in melanoma cells as compared to normal cells. These data are also consistent with similar data from the Human Protein Atlas (Science 347:394). This shows immunohistochemistry for ATP1A1 in sections from normal human skin as well as melanoma specimens (<http://www.proteinatlas.org/ENSG00000163399-ATP1A1/cancer>). Consistent with our results, their data also show that normal skin has very limited ATP1A1 expression while 8 of 10 melanomas have ATP1A1 staining in greater than 75% of cells. We have referenced these data in the revised manuscript.

In Supp Fig 5b, mAtp1a1 only partially rescues the combination treatment phenotype. Is this therapeutic effect because of the MEKi part of the combination?

RESPONSE: That is correct. As shown in Figure 2a and 2d, the growth of M481 tumors is significantly slowed by treatment with MEK inhibitor alone and mouse survival was significantly extended, though not to the same degree as with digoxin plus MEK inhibitor. Since mouse *Atp1a1* only blocks the effect of digoxin, it would be expected to only partially rescue the effects of digoxin plus MEK inhibitor in the experiment shown in Suppl. Fig. 5b.

I find the data in Fig 8b-d a bit confusing. The rationale to specifically choose AML is not clear, and the authors now use cell lines (contrast with the PDX models used in the paper throughout). No mechanistic validation is provided for the AML experiments, which is in contrast to the detailed and elegant work in melanoma.

RESPONSE: The question we addressed in Figure 8 was whether melanomas were uniquely sensitive to digitoxin plus MEK inhibitor or whether other cancers also respond to this drug combination. We focused on *NRAS* mutant AMLs because, like melanoma, they also exhibit MAPK pathway activation, therapy resistance, and poor outcomes (Leukemia 25:1080; Nature 461:441). We had to use cell lines rather than xenografts because we do not have an IRB approval that would allow us to obtain primary patient AMLs. Therefore, we xenografted *NRAS* mutant human AML cell lines into NSG mice and treated with digitoxin and/or MEK inhibitor. We observed that digitoxin and MEK inhibitor exhibited synergistic effects on the survival of mice transplanted with human AML cell lines (Fig. 8c and 8e). These data make the important point that the synergistic effects of this drug combination are not limited to melanoma. Nonetheless, we have not repeated all of the mechanistic studies that were performed in melanoma cells, also in AML, because performing these mechanistic experiments in two different kinds of cancers is beyond what can do in a single manuscript.

I would like to see the HIF1 α data included in the manuscript as an important negative control, both at current dose and at higher dose.

RESPONSE: We have added the requested data (see new Suppl. Fig. 7f). Digitoxin did not inhibit HIF1 α expression in cultured melanoma cells from two patients when used at 50nM, a concentration at the higher end of what we used in vitro and in vivo to observe toxicity against human melanoma cells. In contrast, when we increased digitoxin concentrations to 1250nM, we began to observe reduced HIF1 α expression (see new Suppl. Fig. 7f). Prior studies showed that 100 nM digoxin inhibited HIF1 α expression (Proc. Natl. Acad. Sci. USA 105:19579), well above the range used in our experiments (we used digoxin concentrations of 1-2.5 nM and digitoxin concentrations up to 50nM, similar to the concentrations used in patients). Effects of digitoxin on HIF1 α expression do not appear to explain our results.

Reviewer #3:

This article elaborates on the effects and the potential underlying mechanisms that a combinational use of cardiac glycosides and MEK inhibitor, despite both had been previously investigated individually, to inhibit growth of melanoma cells in vitro and in vivo (xenografted mice) using primary human melanoma cells derived from human patients. The authors first demonstrated that cardiac glycosides preferentially inhibited melanoma cell growth than some normal human cells in vitro (cell culture), which might be related with a high expression level of the ATP1A1 Na⁺/K⁺ pump in melanoma cells. Although no correlation between the ATP1A1 mRNA levels and the IC₅₀ of digitoxin was observed, overexpression of the cardiac glycosides-insensitive mouse ATP1A1 but not the sensitive human ATP1A1 can block the effects of digotoxin on viability and plasmamembrane depolarization in cultured melanoma cells. Then, with their developed xenograft assay, it was clearly shown that a combination of the digotoxin and MEK inhibitor trametinib was significantly more effective, 4 additively and 5-6 synergistically, than either of the single agent alone in 8 of 11 melanoma xenografts derived from patients. The author then investigated the mechanisms to look for causes of the synergistic effects of these two agents. The authors studied the effects of these two types of agents on cell plasma membrane depolarization, ERK phosphorylation in MARK pathway, glucose uptake and metabolism, cell size, intracellular pH, mitochondrial Ca²⁺ level, and mitochondrial function including mitochondrial membrane potentials, ROS level, NADP⁺ and ATP levels. The authors proposed that these two agents synergistically inhibited the Na⁺/H⁺ exchanger (NHE) activity, which led to decrease in intracellular pH, elevation in mitochondrial Ca²⁺ and thus mitochondrial function.

This manuscript involves a large amount of assays and data which appear to be carefully done in most experiments. It is clearly shown that a combinational use of these two agents was more effective in killing melanoma cells, lowering intracellular pH, inhibiting NHE activity, increasing mitochondrial Ca²⁺, and decreasing mitochondrial transmembrane potential, ROS, NAD⁺ and ATP levels. To this reviewer, the main concern of this manuscript is the proposed pivot role of NHE. Although it is safe to draw from the data (Fig. 4d) that digitoxin and MEK inhibitor alone had no effect on NHE activity but together showed strong synergetic effect particularly in M214 cells, the two pieces of evidence to suggest an involvement of NHE1 in intracellular pH and cell death are both weak. The use of a non-specific blocker amiloride to block NHE activity will also block other a family of epithelial Na⁺ channels and other Na⁺ dependent transporter and additionally, the effect was smaller than digitoxin plus MEK inhibitor.

RESPONSE: We believe the data strongly support our conclusion that NHE inhibition contributes to the effects of digoxin plus MEK inhibitor on melanoma cells. As noted in the manuscript, we agree that amiloride inhibits multiple Na⁺ channels and Na⁺-dependent ion transporters and that this by itself does not strongly implicate NHE. But we are not aware of a more specific NHE inhibitor that we could have used in vivo in mice, so more precise pharmacological approaches were not available to us. Moreover, the fact that the effects of amiloride on intracellular pH and melanoma cell death were not as strong as digitoxin plus MEK inhibitor is not interpretable. There are many potential pharmacological reasons why amiloride might be less available in vivo, or a weaker inhibitor of NHE. Without doing extensive pharmacokinetic studies of amiloride in mice it would not be possible to draw any conclusions regarding its capacity to inhibit NHE as compared to digitoxin plus MEK inhibitor.

Secondly, the use of overexpression to rescue only showed very limited success.

RESPONSE: We performed the NHE over-expression experiments because we are not aware of a specific pharmacological inhibitor of NHE that has been used in vivo in mice. We don't believe it is fair to dismiss these results as showing limited success. NHE over-expression significantly but partially rescued the effects of digoxin plus MEK inhibitor on intracellular pH in two different melanomas in vivo (Figure 4g). The lack of a complete rescue might reflect other effects of the drugs or difficult to control technical variables (level of NHE expression, etc.). Importantly, NHE over-expression completely rescued the effects on digoxin plus MEK inhibitor on the survival of M481 cells in vivo while substantially but partially rescuing the effects on M214 cells (Figure 4h). NHE over-expression also significantly rescued the effects of digoxin plus MEKi on mitochondrial Ca²⁺ levels (Figure 5j and 5k). We agree that NHE is not the whole story, and we have noted this in the discussion of the revised manuscript, but to argue that it is not involved one must ignore statistically significant effects in three different assays, using multiple melanomas. Our conclusions are also consistent with multiple prior studies that showed that MEK inhibitor and cardiac glycosides each individually inhibit NHE activity (J. Mol. Cell Cardiol. 20:1; PNAS 109:1239; J. Biol. Chem. 286:13096; J. Biol. Chem. 274:22985).

Additionally, the results from NHE activity assay seems to be inconsistent with those in intracellular pH assays, which seemed to be largely an additive effect of those two agents although the large error bars make it hard to say additive or synergetic.

RESPONSE: It is not formally possible to conclude whether digoxin and MEK inhibitor had additive or synergistic effects on intracellular pH in Figure 4a. We agree that the effects look additive, but they could also be interpreted as synergistic as the individual agents usually had no statistically significant effect on intracellular pH while the combination always did. Figure 4a

shows that the error bars are not large, yet the small effects of the individual agents on intracellular pH generally did not reach statistical significance, and they would not be expected to be biologically significant. Having said this, we agree that digoxin and MEK inhibitor are likely to have cellular effects beyond NHE. We can't define all of the downstream effects in a single study. We can only identify a subset of the key effects. The mechanisms we have identified are statistically significant in many different assays and are sufficient to explain the death of the melanoma cells. We believe they provide meaningful insight into the mechanisms by which these agents work, even if they are not the full story.

Inconsistence exists for M481 in digitoxin-induced effects on intracellular pH (Fig 4a) and NHE activity (Fig 4b). The downstream effects on mitochondrial function were also not included in these tests with amiloride and NHE overexpression. It will be more convincing if digitoxin plus MEK lost their most effects in cell death and mitochondrial function upon a knockdown of NHE1 expression. Therefore, the underlying mechanisms, which were probably more complicated, remained not well established to this reviewer.

RESPONSE: MEK inhibitors have been studied by many labs for more than a decade and have been FDA approved for use in patients for several years and yet the mechanisms downstream of MAPK pathway inhibition are the subject of ongoing studies by many laboratories. Similarly, cardiac glycosides have been studied for several decades and the mechanisms downstream of ATP1A1 that contribute to their clinical effects may not be fully elucidated. No single study can fully elucidate all of the mechanisms by which any drug or drug combination operates.

It is not possible to test whether NHE1 knockdown blocks the effects of digitoxin plus MEK inhibitor on melanoma cells because the NHE family includes 9 members and NHE1, NHE2, NHE3, NHE4, and NHE5 are all thought to function redundantly (Mol Aspects Med 34:236). In addition to NHE1, there are 3 other family members (NHE5, NHE6 and NHE8) that are also expressed by melanoma cells. Digoxin plus MEK inhibitor would be expected to inhibit the function of multiple NHEs. Over-expression of NHE1 is sufficient to partially rescue this effect but knocking down any single NHE would not be expected to phenocopy the effects of digoxin plus MEK inhibitor. To test whether NHEs are necessary for the effects of digoxin plus MEK inhibitor we would have to knock down the expression of multiple NHEs. Given that NHE function is critical for the survival of cancer cells (eLife 2015;4:e03270), this would not only be technically very difficult but the melanoma cells would probably not survive.

The major conclusion in our paper is that ion channel/transporter inhibitors and targeted agents can exhibit synergistic anti-cancer activity. To our knowledge, no other study has ever demonstrated that any ion channel/transporter inhibitor can synergize with any targeted therapy. As indicated by all of the reviewers, the synergistic effects of cardiac glycosides and MAPK pathway inhibitors are well established in our manuscript. Conclusions regarding NHE function are not mentioned in the title or abstract of our manuscript. Nonetheless, we believe the data strongly support the idea that NHE inhibition contributes to the effects of the drug combination.

Specific Comments:

1) additive vs synergetic-- they are clearly different terms; however due to errors/variations presented in the experimental data, sometimes it is hard to definitely call an additive or a synergetic effect of the two agents. For example, in Fig4a, by eye it seems to be an additive effect but statistical analysis may show a synergetic effect which essentially cannot rule out an additive effect because of the large error issue. In Fig2a, the melanoma cells derived from different patients, as the authors stated, some seemed to be additive and some to be synergetic. The manuscript including the title seems to have an emphasis on synergetic effects.

It might be necessary to carefully reexamine the additive and synergetic effects throughout the manuscript, or maybe more appropriate to deemphasize the synergetic effects.

RESPONSE: We have carefully examined the effects throughout the manuscript and believe our use of the terms additive and synergistic is appropriate. We agree that some effects of digitoxin and MEK inhibitor on melanoma cells could be interpreted either as additive or synergistic, based on the magnitude and statistical significance of the effects. When the data did not rigorously distinguish between additive and synergistic effects we concluded that the effects were “additive or synergistic” (e.g. cell death in Fig. 3a; intracellular pH in Fig. 4a; mitochondrial membrane potential in Fig. 6a; ROS levels in Fig. 6d). In other cases, we described the effects as “additive or synergistic” because for some melanoma lines the effects were synergistic (with no effects of individual agents but a significant effect of the combination) while for other melanoma lines the effect could be interpreted as additive. Other effects were consistently synergistic, including the survival of mice with metastatic melanoma (Fig. 2d), mitochondrial calcium levels (Fig. 5b-f), and ATP levels (Fig. 6h). In those cases we described the effects as synergistic. Since survival is the single most important metric of therapy response, it is notable that the drug combination had consistently synergistic effects on the survival of mice with melanomas (Fig. 2d) and leukemias (Fig. 8b-e).

Since we tested the effects of digitoxin plus MEK inhibitor against melanomas obtained from multiple different patients, some differences in response would be expected. We believe it is a strength of our manuscript that we explored the mechanisms in multiple different genetic backgrounds rather than in a single cell line/GEM model. The range of results reported in our manuscript is consistent with the range of results we are observing in our clinical trial. There is no therapy that has uniform effects on all patients. Therefore, we could have performed our experiments in a single background to avoid variability but this would have weakened our study by making it less predictive of the effects expected in the clinic.

All of our major conclusions are based on effects that were consistently observed across most or all of the melanomas we studied: digitoxin combined with MEK inhibitor to synergistically inhibit NHE function (Fig. 4d), leading to intracellular acidification (Fig. 4a), synergistically increased mitochondrial Ca^{2+} levels (Fig. 5b-f), reduced mitochondrial function (Fig. 6a and 6d), synergistically depleted ATP (Fig. 6h), and additively or synergistically increased cell death (Fig. 3a; see summary in Fig. 7a). The net result was synergistically extended survival of mice xenografted with metastatic melanomas (Fig. 2d) and AML (Fig. 8b-e).

Page 10, last sentence of first paragraph: cannot say "does not have additive" because it is kind of additive effect ($A+B = A + B$ although A or $B = 0$).

RESPONSE: If $B=0$ then it might be technically correct to conclude that additive effects are observed even if $A+B=A$. Nonetheless, the common understanding of the term “additive” is that each drug in the combination has a non-zero effect and that the combination reflects the sum of those non-zero effects. If the combination has effects that are indistinguishable from a single agent ($A+B=A$) we don’t think it meaningful to describe the effects as additive. We believe the meaning of the sentence will be more clear to readers as currently written, but will defer to the editors and reviewers if they disagree.

2) The last sentence in the abstract "can synergize with targeted agents" seems to be unclear for the exact meanings.

RESPONSE: Our data indicate that ion transporter inhibitors and targeted agents can exhibit synergistic anti-cancer activity. To our knowledge, no other study has ever demonstrated that any ion channel/transporter inhibitor can synergize with any targeted therapy. We believe this is important because it suggests that ion transporter inhibitors could represent a new class of anti-cancer agents, and that their anti-cancer activity should be assessed in combination with targeted therapies, particularly MAPK pathway inhibitors.

3) It is impressive that a large number of different melanoma cells were used in the initial in vitro (from 15 patients) and in vivo (from 11 patients) cell growth assays. However, in the rest of the manuscript, only M481, M491 and M214 were selectively used. It will be better to explain somewhere why these 3 derived melanoma cells were preferentially used.

RESPONSE: Prior studies have rarely, if ever, examined the mechanisms underlying drug effects by relying almost exclusively on in vivo experiments (rather than in culture), or by using multiple patient-derived xenografts (rather than cell lines). In this way, we believe our manuscript goes well beyond the standards commonly used in this field. Far more time and resources were required to perform our experiments in patient-derived xenografts in vivo rather than in cell lines in culture. For each of our experiments, tumors had to be grown for months in vivo to assess drug effects. Given the time and resources required for such experiments, it is not possible to perform each mechanistic experiment in large numbers of melanomas from many different patients. We performed the mechanistic experiments in M481, M491 and M214 to assess the effects in multiple different genetic backgrounds. The data on these melanomas was not cherry picked from among experiments that involved larger numbers of tumors. We report the results from all of the melanomas that were used in each experiment. These three melanomas were used in the mechanistic experiments because they grow quickly and aggressively in NSG mice. This was convenient for getting the experiments done in a timely way and we believe the data are likely to be more representative of the late stage patients we are treating in clinical trials. Some of the other melanomas that were tested in Figure 2 take much longer to grow, making routine use in mechanistic experiments impractical.

4) Fig 1f, not mentioned in the figure legend.

RESPONSE: Thank you for catching this error. We have expanded the ATP1A1 western blot analysis to include more melanocytes and melanomas. To accommodate the increase in expression data, we have replaced Fig. 1f with a new Supplemental Fig. 1j. We have been careful to describe this panel in the figure legend.

5) Fig 2a, It is a little odd that the M214's data only includes up to 9 days while all others show data extended to 25 days or more. Apparently it is not short of data. In the supplementary Fig 6, similar data of M214 does cover more than 25 days.

RESPONSE: In Figure 2a, the drug treatments were initiated on M214 when the tumors were approximately 1 cm in diameter. At the time this experiment was performed our animal protocol specified that all mice must be euthanized when any mouse in the cohort had a tumor that reached 2 cm in diameter. As a consequence, we were unable to treat M214 cells in the experiment shown in Figure 2a for more than 9 days. Nonetheless, even in this short period we observed clear differences among treatments that were consistent with the conclusions in our manuscript. In Supplemental Fig. 6, the drug treatments were initiated on M214 when the tumors were much smaller, 0.3-0.4 cm in diameter. Also, by the time this experiment was performed our animal protocol had been revised to allow us to wait until tumors reached 2.5 cm before euthanizing the mice. As a result, we could treat these mice with drugs for a longer

period of time. In both experiments, digoxin plus MEKi inhibited the growth of tumors to a significantly greater extent than either single agent alone.

6) *Fig 5e, it might be necessary to explain why FCCP had little effect on the MEK inhibitor + digitoxin -induced increase in mitochondrial Ca²⁺. Additionally, because of the large value (~2000) of time in second, a label in minute might be better for the x-axis in Fig 5a, c-k.*

RESPONSE: We have changed all the x-axes to minutes in Fig. 5a, 5c-k. In most experiments FCCP treatment did reduce mitochondrial calcium levels in melanoma cells treated with digoxin plus MEK inhibitor (see Fig. 5c, 5d, 5f, 5j, and 5k). The only exception was M491 (Fig. 5e). We can't be sure why M491 responded unusually to FCCP treatment. However, other studies have found that there are certain circumstances in which FCCP does not induce Ca²⁺ efflux from mitochondria (see Figure 4B in Cell Metabolism 17:976). It is possible that there is something about the genetic background of this melanoma that causes it to respond unusually to FCCP.

7) *Page 8, last paragraph. It was reported that apoptosis, autophagy and necrosis are involved in melanoma cells death induced by MEK inhibitor. Caspase-3 activation might be not the major cause for melanoma cell apoptosis. It will be helpful to explain why it is chosen as marker of cell death here.*

RESPONSE: Our data suggest that apoptosis is a major mechanism by which digitoxin plus MEK inhibitor induces cell death. Digitoxin plus MEK inhibitor significantly increased the numbers of TUNEL positive nuclei in all patient-derived xenografts (Fig. 3a). Since cells can also become TUNEL positive during necrosis (Mod Pathol 16:389) we also assessed activated caspase-3, a marker of apoptosis (Biochim. Biophys. Acta1833:3448), and observed that digitoxin plus MEK inhibitor also significantly increased the numbers of activated caspase-3⁺ cells in all xenografts (Fig. 3b). Finally, BCL2 over-expression completely rescued the effect of digitoxin plus MEK inhibitor on melanoma cell death, implying an apoptotic mechanism (Fig. 3b).

We have also added new data to the manuscript assessing the effects of digitoxin plus MEK inhibitor on autophagy. We measured p62 and LC3-I/II levels in protein lysates from 3 melanoma xenografts treated with digitoxin and/or MEK inhibitor. We did not detect any consistent effect of digitoxin and/or MEK inhibitor on p62 or LC3-I/II levels (see new Suppl. Fig. 6b). We thus were unable to detect any effect on autophagy. We note in the revised manuscript that it is possible that digitoxin plus MEK inhibitor might influence autophagy or necrosis in some melanomas but that the clearest effects were on apoptosis.

8) *page 9, first line "data not shown". It is better either not to mention the data or actually show the data if mentioned.*

RESPONSE: Our original manuscript has data on numbers of activated caspase-3⁺ cells in melanoma xenografts treated with digitoxin plus MEK inhibitor in Fig. 3b. Therefore, we have revised the paragraph in question to not mention the additional activated caspase-3⁺ data.

9) *Page 10, last sentence of 2nd paragraph is not clear. The point should be there is no good correlation between the digitoxin plus MEK inhibitor -induced cell membrane depolarization, which also happened in non-melanoma cells, and the preferential toxicity of digitoxin plus MEK inhibitor toward melanoma cells.*

RESPONSE: We have revised this sentence as suggested.

10) page 11 and Fig 4a. *The connection between intracellular pH and digitoxin plus MEK inhibitor-induced cell death is weak. The initial pH before treatment (~pH 7.3, 8.0, 7.0) and final pH after treatment (~pH 6.6, 7.2, 6.7) are very different among the used 3 cells, M481, M491 and M214. The difference is also very small before and after treatment in the M214 cells.*

RESPONSE: Intracellular pH is known to be more variable and more alkaline in cancer cells as compared to normal cells (Nature Reviews Cancer 5:786). Thus, our results appear to reflect real biological heterogeneity among tumors. Despite this heterogeneity, treatment with digitoxin plus MEK inhibitor significantly reduced intracellular pH in all three melanomas. Moreover, the differences would be expected to be biologically significant. Digitoxin plus MEK inhibitor reduced intracellular pH by 0.7 units in M481, 0.9 units in M491 and 0.3 units in M214 (Fig. 4a). A 0.3 unit decrease in intracellular pH is sufficient to induce apoptosis in some cells, including cancer cells (see Figures 2f, 3f and 5c in Nature Cell Biology 2:318; see Figure 9 in eLife 2015; 4:e03270). Therefore, the declines in intracellular pH in our experiments would be expected to induce cell death, just as we observed. Our ability to partially or completely rescue the effects of digitoxin plus MEK inhibitor on intracellular pH (Figure 4g) and cell death (Figure 4h) by NHE1 over-expression further supports our conclusions.

11) page 13. *"Intracellular acidification increases intracellular Ca²⁺ levels by inhibiting various Ca²⁺ channels". It is better to correctly use the term "channel" and transporter". The former passively moves Ca²⁺ across the Ca²⁺ gradient, thus will decrease the intracellular Ca²⁺ after inhibition. The authors may mean transporters here.*

RESPONSE: Thank you, we have corrected this.

The evidence for connection between intracellular pH decrease and mitochondrial Ca²⁺ elevation seems to be not strong here.

RESPONSE: As described in the second paragraph of the discussion, a number of published studies suggest that decreased intracellular pH can increase intracellular Ca²⁺ (J. Physiol. 398:391; Am. J. Physiol. 264:4448; Am. J. Physiol. 261:617). Most of these studies did not distinguish between cytoplasmic versus mitochondrial Ca²⁺. Therefore, it is hard to compare our findings to these studies. Nonetheless, we found that digitoxin plus MEK inhibitor reduced intracellular pH (Fig. 4a) and elevated mitochondrial Ca²⁺ levels (Fig. 5b-f). Over-expression of NHE1 partially rescued the effects of digitoxin plus MEK inhibitor on intracellular pH (Fig. 4g), and mitochondrial Ca²⁺ levels (Fig. 5j and 5k), and partially or completely rescued the effects on cell death (Fig. 4h). The evidence does, therefore, suggest a strong link between intracellular pH and mitochondrial Ca²⁺ levels.

12) page 14. *The use of BAPTA-AM is problematic and the data is hard to interpret because of its non-specific effects on all cellular Ca²⁺.*

RESPONSE: We agree that BAPTA-AM is not ideal, but we have not been able to identify any other calcium chelator that is not toxic to melanoma cells. This experiment was designed to test whether calcium fluxes induced by digitoxin plus MEK inhibitor contribute to the induction of cell death. We tested whether BAPTA-AM could reduce cell death induced by digitoxin plus MEK inhibitor in melanoma xenografts. Interpretation of this experiment is complicated by the fact that BAPTA-AM itself induced cell death in most melanomas to an extent similar to digitoxin plus MEK inhibitor, rendering the data uninterpretable in these melanomas. When we treated M214 xenografts with digitoxin plus MEK inhibitor, BAPTA-AM was reasonably well tolerated by the cells and it did appear to rescue the effect of digitoxin plus MEK inhibitor on cell death (Fig. 5l).

These results are consistent with the idea that calcium dysregulation in the presence of digitoxin plus MEK inhibitor contributes to the induction of cell death in melanomas, though we have not been able to identify a pharmacological modulator of calcium levels that is non-toxic and that would allow us to test this in additional melanomas. We have discussed the limitations of BAPTA-AM in the original manuscript (page 14, second paragraph). In any case, none of the major conclusions in the manuscript depend on the BAPTA-AM data. We could remove the data, but we think it better to leave them in, even if the data are limited in scope.

Reviewer #4:

The manuscript by Eskiocak U et al. investigated the efficacy of the combination of cardiac glycosides and MAPK inhibitors in treating melanoma in vitro and in vivo. The most difficult hurdle in the context of BRAF-targeted therapies is to overcome the acquisition of therapy resistance to MAPK inhibitors. The authors did not investigate the combination therapy presented in this manuscript toward that end.

RESPONSE: This was not the point of our manuscript. Many prior studies have studied the mechanisms by which BRAF mutant melanomas acquire resistance to BRAF or MEK inhibitors. The point of our manuscript was to assess a new drug combination, cardiac glycoside plus MEK inhibitor, and the mechanisms by which it has synergistic effects on the survival of mice xenografted with human melanomas. One surprising observation was that digitoxin plus MEK inhibitor exhibited strong activity against even BRAF wild-type melanomas (Fig. 2a). Given that BRAF wild-type melanomas generally do not respond clinically to trametinib (Lancet Oncol. 13:782), the pre-clinical data in our manuscript led to the initiation of a Phase 1b clinical trial of digoxin plus trametinib in patients with advanced, refractory BRAF wild-type melanoma (clinicaltrials.gov #NCT02138292). The bottom line is that mechanisms of therapy resistance in BRAF mutant melanomas is an interesting and important question, but it is a completely different question than the one we set out to address in our pre-clinical and clinical studies. It is equally important to improve therapies for patients with BRAF wild-type melanomas as to defeat resistance mechanisms in BRAF mutant melanomas.

Major comments are:

1. The reviewer is not convinced by the rationale of investigating the efficacy of cardiac glycosides in malignant melanoma, which has not been fully addressed.

RESPONSE: Our rationale for studying the effects of cardiac glycosides in melanoma arose from a series of small molecule screens in which over 200,000 small molecules were screened for preferential toxicity against primary human melanoma cells. We screened for small molecules that exhibited significantly more toxicity against melanoma cells than against normal human melanocytes. All of the cardiac glycosides in the libraries came up as hits in the screens. This persuaded us that melanoma cells are unusually sensitive to cardiac glycosides and motivated us to perform the experiments described in our manuscript. We have revised the beginning of the results section of our manuscript to make this more clear.

2. Please strengthen data shown in Fig 1e by analyzing gene expression microarray data included in GSE46517 and TCGA SKCM. Break down samples into skin melanocytes, nevi, primary melanoma and metastatic melanoma (GSE46517) or primary melanoma and metastatic melanoma (TCGA SKCM).

RESPONSE: We extracted *ATP1A1* expression from GEO data sets (GDS1375 and GSE46517) and compared the expression levels in normal skin, benign nevi, primary melanoma, and malignant melanoma (see new Supplementary Fig. 1g-h). Consistent with the

conclusions in our original manuscript, both datasets showed that ATP1A1 expression levels were significantly higher in melanomas as compared to normal skin. There was a trend toward higher expression in melanoma as compared to benign nevi in both data sets, but the effect was statistically significant in only one of the datasets. These published data are consistent with the conclusions in our paper.

Also consistent with the conclusions in our paper, the GSE46517 data described above show that ATP1A1 expression did not significantly differ between primary and metastatic melanomas (see new Suppl. Fig. 1h). The Cancer Genome Atlas (TCGA) RNAseq data (<https://tcga-data.nci.nih.gov/tcga>) also show that there is no difference in ATP1A1 expression between primary and metastatic melanoma. To avoid redundancy we have not reprinted the publicly available TCGA data but we note these data in the text of our manuscript.

3. Please strengthen data shown in Fig 1e and 1f by performing IHC staining of ATP1A1 in normal skin, nevi, primary melanoma and metastatic melanoma.

RESPONSE: We have added new data showing immunohistochemical staining for ATP1A1 in normal human skin, benign nevus, primary melanoma, and metastatic melanoma specimens (see new Suppl. Fig. 1i). In normal skin, ATP1A1 was limited to basal keratinocytes but in benign nevi, ATP1A1 staining expanded to include melanocytic nests (Suppl. Fig. 1i). In primary and metastatic melanomas, ATP1A1 staining was robust and nearly homogeneous among melanoma cells (Suppl. Fig. 1i). Since the western data was collected from dissected tumors, which included variable amounts of melanoma cells and normal stromal cells, these IHC data more definitively demonstrate that ATP1A1 expression tends to increase in melanoma cells as compared to normal cells. These data are also consistent with similar data from the Human Protein Atlas (Science 347:394; <http://www.proteinatlas.org/ENSG00000163399-ATP1A1/cancer>). Consistent with our results, their data also show that normal skin has very limited ATP1A1 expression while 8 of 10 melanomas have ATP1A1 staining in greater than 75% of cells. We have referenced these data in the revised manuscript.

4. Fig 1g - ATP1A1 was expressed at a higher level in melanomas compared to that in melanocytes. Why not over expressing hATP1A1 in hMEL1 and hMEL2 and then treating them with Digitoxin?

RESPONSE: First, we should note that we wrote in the original manuscript that “We observed no correlation between ATP1A1 mRNA levels and digitoxin IC₅₀ values (Supplementary Fig. 1k)”. Therefore, we have already noted that the increased sensitivity of melanoma cells to cardiac glycosides is not driven only by their increased expression of ATP1A1. Therefore, we didn’t perform the hATP1A1 over-expression experiment in normal melanocytes because our data indicate that this would not be predicted to be sufficient to increase their sensitivity to cardiac glycosides. In response to this suggestion, we have now attempted to perform this experiment but low infection efficiency in normal melanocytes prevented us from testing whether hATP1A1 over expressing normal melanocytes would be more sensitive to digitoxin. Since these are primary normal cells that grow very slowly in culture it is very difficult to expand the infected cells for this experiment. Given that we already note in the manuscript that there is no correlation between ATP1A1 expression and digitoxin IC₅₀ values, we don’t believe that results from the ATP1A1 over-expression experiment would change any of our conclusions.

5. It is absolutely critical that the authors perform the knock-down experiment in M481, M491 and M214 to demonstrate if siATP1A1 or shATP1A1 can recapitulate the efficacy of Digitoxin.

RESPONSE: To address this question we transfected three different siRNAs against *ATP1A1* into A375 melanoma cells as well as M481 and M214 melanomas. We used siRNA against an essential gene, ubiquitin B (UBB), as a positive control. In A375 cells, each of the siATP1A1s knocked down *ATP1A1* expression and killed the melanoma cells, consistent with our conclusions (see new Suppl. Fig. 1c-d and data below). However, the siRNAs were not effective at knocking down *ATP1A1* expression in any of the primary melanoma cells (see data below). There are a number of possibilities for why the siRNAs were effective at knocking down *ATP1A1* in A375 cells but not in primary melanoma cells. Absent the ability to consistently and predictably knockdown *ATP1A1* in primary melanoma cells we are unable to do this experiment in primary melanomas. Nonetheless, the data from A375 cells support our conclusions.

The data in our manuscript strongly support our conclusion that the effects of digitoxin on melanoma cells are mediated by *ATP1A1* because over expression of mouse *ATP1A1* (which binds cardiac glycosides poorly and therefore is insensitive to them) completely rescues the effects of cardiac glycosides on melanoma cells (Fig. 1e-f). Multiple previously published studies have used over-expression of mouse *ATP1A1* to rescue the effects of cardiac glycosides as a way of showing that the effects were on-target (Nature 472:486; Nature Chemical Biology 7:29; PLoS One 4:e8292).

6. Fig 1g and 1h - the comparison made in this panel did not make sense. For example, the authors should have compared the control cells to cells over-expressing *mAtp1a1* or *hATP1A1* when treated with DMSO or Digitoxin at 25 or 50nM.

RESPONSE: We have revised these figures to compare all treatments to untreated control cells. The revised data presentation continues to support the conclusions in the original manuscript: over-expression of *mAtp1a1*, but not *hATP1A1*, rescued the effects of digitoxin on cell viability and membrane potential in melanoma cells from all three patients (Fig. 1e-f).

7. Fig 2a - please replace "Diameter (cm)" with "Tumor Volumes (mm³)". Basically, Digitoxin as a single agent did not work in most cases, which is not in line with in vitro data shown in Fig 1.

RESPONSE: We are unable to replace the diameter data in Figure 2a with tumor volume data

because we did not collect data on tumor volume. Nonetheless, the tumor diameter data in Figure 2a do reflect the actual effects of digitoxin in vivo. As noted in our results, and as explained in the discussion of our paper (see the opening sentence of the discussion on page 18), cardiac glycosides as single agents had marginal activity in vivo (Fig. 2a and 2d; Suppl. Fig. 3) despite the activity we observed in culture (Fig. 1a-c). In addition to having little effect on the diameters of most melanomas in vivo, there also wasn't much effect on the survival of mice with metastatic melanoma (Fig. 2d). Our results may explain why, as far as we can tell, none of the prior clinical trials that tested cardiac glycosides for anti-cancer activity have ever reported results. Unfortunately, it is not uncommon in cancer biology to find drugs that exhibit promising activity in culture but little activity in vivo. For this reason, we put enormous effort into testing cardiac glycosides, with or without MAPK pathway inhibitors, in xenografts in vivo. Our experiments showed that cardiac glycosides synergize with MAPK pathway inhibitors to extend the survival of mice with metastatic melanoma. To our knowledge, this is the first indication that any ion transporter inhibitor can synergize with a targeted agent. The results of our ongoing clinical trial support the conclusion that these agents also synergize in a subset of patients.

8. Fig 3d, 3f, 3g, 3h, 3i - MEKi or Digitoxin alone were missing. This shall also apply to the remaining figures when data of the single inhibitor control be included.

RESPONSE: The experiments in Fig 3d, 3f, 3g, 3h, and 3i were designed to test whether the combination of digitoxin plus MEK inhibitor kills normal human cells. This was important to get an indication of whether the drug combination might pose safety concerns in the clinical trial. There was no point in testing MEK inhibitor alone or digitoxin alone in these experiments because there is already a long clinical history of using MEK inhibitor alone or digitoxin alone in patients. We already know that when used within the therapeutic range employed in our experiments that as single agents these drugs don't kill normal cells and that they are tolerated by patients. Thus, including these as single agents in the experiments in Figure 3 would not have added anything beyond the large bodies of pre-clinical and clinical data that are already available for these drugs.

9. Fig 3j - data basically showed that the decrease in pERK when cells were treated with the combo was due to the treatment of MEKi, which did not support the additive effect or synergy between MEKi and Digitoxin. A more in-depth understanding of why the combo was working better than either single agent is clearly warranted in this study.

RESPONSE: Yes, that is exactly what we concluded, that there is no additive or synergistic effect of digitoxin plus MEK inhibitor at the level of MAPK pathway activation - that the additive and synergistic effects arise downstream of the MAPK pathway. Figures 4-6 address the question of why the drug combination exhibits synergistic effects on the survival of mice with metastatic melanoma. As summarized in Figure 7a, our data suggest that digitoxin plus cardiac glycosides synergistically inhibit NHE function (Figure 4d), leading to intracellular acidification (Fig. 4a), synergistically increased mitochondrial Ca^{2+} levels (Fig. 5b-f), reduced mitochondrial function (Fig. 6a and 6d), synergistically depleted ATP (Fig. 6h), and additively or synergistically increased cell death (Fig. 3a; see summary in Fig. 7a).

10. Fig 8 - Digoxin single agent was missing for M481, M491 and M610.

RESPONSE: The effect of digitoxin as a single agent against M481, M491, and M610 was already shown in Figure 2a. In M481 and M491 there was no significant effect. In M610 the effect was significant but modest. Given that we knew single agent activity against these melanomas was modest, we did not include cardiac glycoside as a single agent in the

experiments shown in Figure 8. The experiments shown in Figure 8 were already very complex, involving large numbers of mice and many treatments. Since the point of these experiments was to test whether cardiac glycoside plus MEKi plus BRAFi had significantly more activity against melanomas than MEKi plus BRAFi, it was not necessary to test single agent activity of cardiac glycosides against these melanomas for a second time.

11. The authors used "Additively" and "Synergistically" throughout this manuscript. Be very careful about using these words if no scientific evidence are in place to support this claim. For the in vitro studies, the authors would have diluted either single agent in order to calculate Bliss values. The situation is more complicated for interpreting in vivo data. Please consider using "Significantly" or "Markedly".

RESPONSE: We believe our use of the terms is always consistent with the data. Some effects of digitoxin and MEK inhibitor on melanoma cells could be interpreted either as additive or synergistic, based on the magnitude and statistical significance of the effects. When the data did not rigorously distinguish between additive and synergistic effects we concluded that the effects were “additive or synergistic” (e.g. cell death in Fig. 3a; intracellular pH in Fig. 4a; mitochondrial membrane potential in Fig. 6a; ROS levels in Fig. 6d). In other cases, we described the effects as “additive or synergistic” because for some melanoma lines the effects were synergistic (with no effects of individual agents but a significant effect of the combination) while for other melanoma lines the effect could be interpreted as additive. Other effects were consistently synergistic, including the survival of mice with metastatic melanoma (Fig. 2d), mitochondrial calcium levels (Fig. 5b-f), and ATP levels (Fig. 6h). In those cases we described the effects as synergistic. Since survival is the single most important metric of therapy response, it is notable that the drug combination had consistently synergistic effects on the survival of mice with melanomas (Fig. 2d) and leukemias (Fig. 8b-e).

We hope these revisions will render our manuscript acceptable for publication.

Sincerely,

Sean J. Morrison
Investigator, Howard Hughes Medical Institute
Director, Children’s Research Institute
University of Texas Southwestern Medical Center

Reviewers' comments:

Reviewer #1 (Remarks to the Author):

I am satisfied with the investigators's responses.

Reviewer #2 (Remarks to the Author):

The authors have adequately addressed the questions that were raised during review. As such, I have no further comments and would recommend publication of this article.

Reviewer #3 (Remarks to the Author):

This revised manuscript has addressed many concerns from the reviewers. If unaddressed, the authors explained or argued for the reasons. It is understandable that it is hard to address all the concerns raised by the reviewers. The underlying mechanisms of the combinational effects of cardiac glycosides and MEK inhibitor can be complex and unlikely solved in a single study. To this reviewer, the authors have provided useful clinically relevant findings and also made some significant progress to understand the underlying mechanisms. The revised manuscript is suitable for publication.

Reviewer #4 (Remarks to the Author):

This revised manuscript has been somewhat improved. Some comments were not adequately addressed. The reviewer still had issues with this study (outlined below) and would like the authors to obtain an in-depth understanding of molecular mechanisms underlying how digitoxin and MEK inhibitors work downstream of MAPK signaling pathway.

1. In 7 PDX models tested in Fig 2a which did not harbor BRAF V600E mutation, digitoxin alone was also effective in impairing tumor growth exhibited by three of them (M405, M214 and M715), whereas MEK inhibitor alone was effective in impairing tumor growth exhibited by all of them except M711. One could argue that MEK inhibitor plus another targeted agent will achieve a greater inhibitory effect. The reviewer is not convinced by this conclusion suggesting the efficacy of digitoxin plus MEKi for treating BRAF wild-type melanomas.
2. How many samples the authors used to carry out ATP1A1 IHC staining in Fig S1i? Please quantify and score the staining.
3. The reviewer is concerned about the fact that data on tumor volumes were not collected throughout all in vivo experiments. The alternative approach is to quantify bioluminescence signals.
4. The authors argue that the additive and synergistic effects occur downstream of MAPK pathway without showing any evidence to prove this point. The reviewer would like to see more mechanistic insights into the combinatorial effect of these two agents.
5. It seems that the authors use "additively" and "synergistically" interchangeably throughout this manuscript. These terms should be clearly defined in this study.

Reviewer #1:

I am satisfied with the investigators's responses.

RESPONSE: Thank you

Reviewer #2:

The authors have adequately addressed the questions that were raised during review. As such, I have no further comments and would recommend publication of this article.

RESPONSE: Thank you

Reviewer #3:

This revised manuscript has addressed many concerns from the reviewers. If unaddressed, the authors explained or argued for the reasons. It is understandable that it is hard to address all the concerns raised by the reviewers. The underlying mechanisms of the combinational effects of cardiac glycosides and MEK inhibitor can be complex and unlikely solved in a single study. To this reviewer, the authors have provided useful clinically relevant findings and also made some significant progress to understand the underlying mechanisms. The revised manuscript is suitable for publication.

RESPONSE: Thank you

Reviewer #4:

This revised manuscript has been somewhat improved. Some comments were not adequately addressed. The reviewer still had issues with this study (outlined below) and would like the authors to obtain an in-depth understanding of molecular mechanisms underlying how digitoxin and MEK inhibitors work downstream of MAPK signaling pathway.

1. In 7 PDX models tested in Fig 2a which did not harbor BRAF V600E mutation, digitoxin alone was also effective in impairing tumor growth exhibited by three of them (M405, M214 and M715), whereas MEK inhibitor alone was effective in impairing tumor growth exhibited by all of them except M711. One could argue that MEK inhibitor plus another targeted agent will achieve a greater inhibitory effect. The reviewer is not convinced by this conclusion suggesting the efficacy of digitoxin plus MEKi for treating BRAF wild-type melanomas.

RESPONSE: Of course it is possible that a different drug combination could be discovered in future that is more effective than digitoxin plus MEK inhibitor. Nonetheless, no such combination has yet been discovered for BRAF wild-type melanomas. Moreover, even if we spend years testing every possible combination of known drugs to compare their efficacy versus digitoxin plus MEK inhibitor the results would not affect any of the conclusions in our manuscript. The major conclusion in our paper is that ion transporter inhibitors and targeted agents can exhibit synergistic anti-cancer activity. This is shown in many figures within our manuscript including Figure 2a where digitoxin plus MEK inhibitor exhibited synergistic activity against at least 4 of 11 melanomas (M481, M634, M514, M660; three of which are BRAF wild-type). In each case, digitoxin had little or no effect on its own but digitoxin plus MEK inhibitor had effects that were significantly greater than MEK inhibitor alone. Digitoxin plus MEK inhibitor also synergistically extended the survival of mice with metastatic melanoma (Figure 2d). To our knowledge, no other study has demonstrated that any ion channel/transporter inhibitor can synergize with any targeted therapy. We believe this is conceptually important because it suggests that ion transporter inhibitors could represent a new class of anti-cancer agents and that their anti-cancer activity should be assessed in combination with targeted therapies, particularly MAPK

pathway inhibitors. These conclusions will remain true even if it is possible to identify combinations of drugs that are more effective than digitoxin plus MEK inhibitor in future.

Given that BRAF wild-type melanomas generally do not respond clinically to trametinib (Lancet Oncol. 13:782), the pre-clinical data in our manuscript led to the initiation of a Phase 1b clinical trial of digoxin plus trametinib in patients with advanced, refractory BRAF wild-type melanoma (clinicaltrials.gov #NCT02138292). This clinical trial could not have gone forward if any of the physicians involved in the study, or any of the committees that approved our study, were aware of any combination of targeted therapies that show increased evidence of efficacy relative to digoxin plus MEK inhibitor.

2. How many samples the authors used to carry out ATP1A1 IHC staining in Fig S1i? Please quantify and score the staining.

RESPONSE: We examined the expression of ATP1A1 in normal skin, benign nevi, primary melanoma, and metastatic melanoma sections using 3 or 4 independent samples per lesion type. A dermatopathologist and a dermatologist each scored the melanocytic portions of each sample as having none, low, medium, or high intensity staining of ATP1A1. The findings are tabulated below and are consistent with our conclusion that ATP1A1 is robustly expressed in melanoma cells, at somewhat higher levels than in melanocytes. Both primary and metastatic melanomas expressed ATP1A1 robustly (representative images in Supp. Fig. 1i). We will add these data to the manuscript. However, it should be noted that these data are consistent with several other types of data that are shown or referenced in our manuscript, including independently published work, that shows that ATP1A1 is robustly expressed by melanoma cells, and that the expression levels tend to be higher than in normal melanocytes. In particular,

SAMPLE	ATP1A1 STAINING INTENSITY (none, low, medium, or high)
NORMAL SKIN	
TC16_1666	Medium
TC16_1783	Medium
TC16_1820	Medium
BENIGN NEVI:	
TC16_161	High
TC16_161B	Medium
TC16_834	High
TC16_1757	High
PRIMARY MELANOMA:	
TC15_7226	High
TC15_6444	High
TC16_1335	Medium
METASTATIC MELANOMA:	
TC15_4648	High
TC15_4664A	High
TC15_4664B	Medium

our data are consistent with published immunohistochemistry data for ATP1A1 in the Human Protein Atlas (see p.5, lines 117-119 in our manuscript). We have also quantified ATP1A1 expression in melanocytes and melanomas by Western blot (Supp. Fig. 1j), by microarray analysis of 33 melanomas and 3 melanocytes (Fig. 1e, Supp. Fig. 1e), and by published microarray data of normal skin, benign nevi, and melanoma samples (Fig. 1g and 1h). Our results, as well as the published results from others, show that ATP1A1 is robustly expressed by all melanomas and at lower levels in normal melanocytes and normal skin. ATP1A1 expression in nevi and immortalized melanocytes is variable, sometimes similar to melanoma cells.

3. The reviewer is concerned about the fact that data on tumor volumes were not collected throughout all in vivo experiments. The alternative approach is to quantify bioluminescence signals.

RESPONSE: Tumor diameters are used very commonly as a measure of treatment response (e.g. see recent studies in Nature 515:578, Figures 1 and 2; Cell 162:1230 Figs. 1, 3, 4; Blood 100:4051, Fig. 1). In fact, the RECIST criteria that are THE standard for assessing clinical treatment responses are entirely based on changes in tumor diameter, not tumor volume, even though tumor volumes could be calculated from the same CT images that are used to determine tumor diameters (Eur. J. Cancer 45:228). Thus, our use of tumor diameter measurements adheres to the most widely accepted methods as well as the methods that have been used in our clinical trial of digitoxin plus MEK inhibitor in patients.

Photography of tumors at the end of some experiments showed that the effects of digitoxin plus MEK inhibitor on tumor diameter correlated with the effects on tumor volume, as would be expected (see Suppl Fig. 3). Thus, there is every reason to believe that our conclusions would not change if based on tumor volume instead of tumor diameter.

We also used several different metrics to show synergistic effects of digitoxin plus MEK inhibitor on tumor size. These include tumor diameter (Fig. 2a), mouse survival (Fig. 2d), bioluminescence imaging (Suppl. Fig. 4), photography of tumors (Suppl. Fig. 3), and frequency of melanoma cells undergoing cell death (Figure 3a). In every case the data support the same conclusion: that digitoxin and MEK inhibitor have synergistic effects on melanomas from a subset of patients, irrespective of how tumor burden was measured.

4. The authors argue that the additive and synergistic effects occur downstream of MAPK pathway without showing any evidence to prove this point. The reviewer would like to see more mechanistic insights into the combinatorial effect of these two agents.

RESPONSE: The manuscript includes extensive mechanistic data as summarized in Figure 7a.

5. It seems that the authors use "additively" and "synergistically" interchangeably throughout this manuscript. These terms should be clearly defined in this study.

RESPONSE: This is not true, as explained in detail in our response to the same question in the last round of review (point #11 from reviewer #4; pasted below for your convenience).

11. The authors used "Additively" and "Synergistically" throughout this manuscript. Be very careful about using these words if no scientific evidence are in place to support this claim. For the in vitro studies, the authors would have diluted either single agent in order to calculate Bliss values. The situation is more complicated for interpreting in vivo data. Please consider using "Significantly" or "Markedly".

RESPONSE: We believe our use of the terms is always consistent with the data. Some effects of digitoxin and MEK inhibitor on melanoma cells could be interpreted either as additive or synergistic, based on the magnitude and statistical significance of the effects. When the data did not rigorously distinguish between additive and synergistic effects we concluded that the effects were "additive or synergistic" (e.g. cell death in Fig. 3a; intracellular pH in Fig. 4a; mitochondrial membrane potential in Fig. 6a; ROS levels in Fig. 6d). In other cases, we described the effects as "additive or synergistic" because for some melanoma lines the effects were synergistic (with no effects of individual agents but a significant effect of the combination) while for other melanoma lines the effect could be interpreted as additive. Other effects were consistently synergistic, including the survival of mice with metastatic melanoma (Fig. 2d), mitochondrial calcium levels (Fig. 5b-f), and ATP levels (Fig. 6h). In those cases we described the effects as synergistic. Since survival is the single most important metric of therapy response, it is notable that the drug combination had consistently synergistic effects on the survival of mice with melanomas (Fig. 2d) and leukemias (Fig. 8b-e).

We hope these clarifications will render our manuscript acceptable for publication.

Sincerely,

Sean J. Morrison
Investigator, Howard Hughes Medical Institute
Director, Children's Research Institute
University of Texas Southwestern Medical Center